# Multiscale analysis of single and double maternal-zygotic *Myh9* and *Myh10* mutants during mouse preimplantation development

**Markus Frederik Schliffka[1,2†], Anna Francesca Tortorelli[1†], Özge Özgüç[1], Ludmilla de Plater[1], Oliver Polzer[1], Diane Pelzer[1], Jean-Léon Maître[1]\***

[1]Institut Curie, PSL Research University, Sorbonne Université, Paris, France; [2]Carl Zeiss SAS, Marly-le-Roy, France

**Abstract** During the first days of mammalian development, the embryo forms the blastocyst, the structure responsible for implanting the mammalian embryo. Consisting of an epithelium enveloping the pluripotent inner cell mass and a fluid-filled lumen, the blastocyst results from a series of cleavage divisions, morphogenetic movements, and lineage specification. Recent studies have identified the essential role of actomyosin contractility in driving cytokinesis, morphogenesis, and fate specification, leading to the formation of the blastocyst. However, the preimplantation development of contractility mutants has not been characterized. Here, we generated single and double maternal-zygotic mutants of non-muscle myosin II heavy chains (NMHCs) to characterize them with multiscale imaging. We found that *Myh9* (NMHC II-A) is the major NMHC during preimplantation development as its maternal-zygotic loss causes failed cytokinesis, increased duration of the cell cycle, weaker embryo compaction, and reduced differentiation, whereas *Myh10* (NMHC II-B) maternal-zygotic loss is much less severe. Double maternal-zygotic mutants for *Myh9* and *Myh10* show a much stronger phenotype, failing most of the attempts of cytokinesis. We found that morphogenesis and fate specification are affected but nevertheless carry on in a timely fashion, regardless of the impact of the mutations on cell number. Strikingly, even when all cell divisions fail, the resulting single-celled embryo can initiate trophectoderm differentiation and lumen formation by accumulating fluid in increasingly large vacuoles. Therefore, contractility mutants reveal that fluid accumulation is a cell-autonomous process and that the preimplantation program carries on independently of successful cell division.

**\*For correspondence:**
jean-leon.maitre@curie.fr

[†]These authors contributed equally to this work

## Introduction

During embryonic development, cells execute their genetic program to build organisms with the correct cell fate, shape, position, and number. In this process, the coordination between cell proliferation, differentiation, and morphogenesis is crucial. In mice, the early blastocyst initially consists of 32 cells distributed among the trophectoderm (TE) and inner cell mass (ICM), with squamous TE cells enveloping the ICM and a lumen called the blastocoel (*Frankenberg et al., 2016*; *Płusa and Piliszek, 2020*; *Rossant, 2016*). Starting from the zygote, the early blastocyst forms after five cleavage divisions, three morphogenetic movements, and two lineage commitments (*Maître, 2017*; *White et al., 2018*; *Zhang and Hiiragi, 2018*). Differentiation and morphogenesis are coupled by the formation of a domain of apical material after the third cleavage (*Ziomek and Johnson, 1980*). The apical domain promotes differentiation into TE by counteracting signals from cell-cell contacts, which control the nuclear localization of the co-transcriptional activator Yap (*Hirate et al., 2013*; *Nishioka et al., 2009*; *Wicklow et al., 2014*). The apical domain also guides cell division orientation

(*Dard et al., 2009*; *Korotkevich et al., 2017*; *Niwayama et al., 2019*) and contact rearrangements (*Maître et al., 2016*), which is key for positioning cells at the embryo surface or interior. Morphogenesis and cleavages are concomitant and appear synchronized with compaction starting after the third cleavage, internalization after the fourth, and lumen opening after the fifth. Beyond this apparent coordination, the existence of a coupling between cleavages and morphogenesis requires further investigations.

Actomyosin contractility is a conserved instrument driving animal morphogenesis (*Heisenberg and Bellaïche, 2013*; *Lecuit and Yap, 2015*) and cytokinesis (*Ramkumar and Baum, 2016*). Recent studies have suggested key contributions of actomyosin contractility during all the morphogenetic steps leading to the formation of the blastocyst (*Özgüç and Maître, 2020*). During the 8-cell stage, increased contractility at the cell-medium interface pulls blastomeres together and compacts the embryo (*Maître et al., 2015*). Also, cells form an apical domain that inhibits actomyosin contractility (*Maître et al., 2016*; *Zhu et al., 2017*). During the fourth cleavage, the asymmetric inheritance of this domain leads sister cells to exhibit distinct contractility (*Anani et al., 2014*; *Maître et al., 2016*). This causes the most contractile blastomeres to internalize and adopt ICM fate, while weakly contractile cells are stretched at the surface of the embryo and become TE (*Maître et al., 2016*; *Samarage et al., 2015*). When the blastocoel fluid starts to accumulate, these differences in contractility between ICM and TE cells guide the fluid away from ICM-ICM cell-cell contacts (*Dumortier et al., 2019*). Finally, contractility has been proposed to control the size of the blastocoel negatively by increasing the tension of the TE and positively by reinforcing cell-cell adhesion via mechanosensing (*Chan et al., 2019*). Together, these findings highlight actomyosin as a major engine powering blastocyst morphogenesis.

To change the shape of animal cells, myosin motor proteins contract a network of cross-linked actin filaments, which can be tethered to the plasma membrane, adherens junctions, and/or focal adhesions (*Murrell et al., 2015*). This generates tension, which cells use to change shape and tissue topology (*Clark et al., 2013*; *Salbreux et al., 2012*). Among myosin motors, non-muscle myosin II are the key drivers of cell shape changes (*Zaidel-Bar et al., 2015*). Non-muscle myosin II complexes assemble from myosin regulatory light chains, myosin essential light chains, and non-muscle myosin heavy chains (NMHCs) (*Vicente-Manzanares et al., 2009*). NMHCs are responsible for generating the power stroke and exist in three distinct paralogs in mice and humans: NMHC II-A, II-B, and II-C, encoded by the genes *Myh9/MYH9*, *Myh10/MYH10,* and *Myh14/MYH14* (*Conti et al., 2004*). Distinct paralogs co-assemble into the same myosin mini-filaments (*Beach et al., 2014*) and, to some extent, seem to have redundant actions within the cells. However, several in vitro studies point to specific roles of NMHC paralogs. For example, MYH9 plays a key role in setting the speed of furrow ingression during cytokinesis (*Taneja et al., 2020*; *Yamamoto et al., 2019*) and is essential to drive bleb retraction (*Taneja and Burnette, 2019*). During cell-cell contact formation, MYH9 was found essential for cadherin adhesion molecule clustering and setting contact size while MYH10 would be involved in force transmissions across the junction and would influence contact rearrangements (*Heuzé et al., 2019*; *Smutny et al., 2010*).

These studies at the subcellular level and at a short timescale complement those at the organismal level and at a long timescale. In mice, the zygotic knockout of *Myh14* causes no obvious phenotype with animals surviving to adulthood with no apparent defect (*Ma et al., 2010*), whereas the loss of either *Myh9* (*Conti et al., 2004*) or *Myh10* (*Tullio et al., 1997*) is embryonic lethal. *Myh9* zygotic knockout embryos die at E7.5 due to visceral endoderm adhesion defects (*Conti et al., 2004*). *Myh10* zygotic knockout mice die between E14.5 and P1 because of heart, brain, and liver defects (*Tullio et al., 1997*). In addition, knocking out both *Myh10* and *Myh14* can lead to abnormal cytokinesis (*Ma et al., 2010*). Elegant gene replacement experiments have also revealed insights into partial functional redundancy between *Myh9* and *Myh10* during development (*Wang et al., 2010*). However, despite the prominent role of actomyosin contractility during preimplantation development (*Ma et al., 2010*), the specific functions of NMHC paralogs remain largely unknown. Previous genetic studies have relied on zygotic knockouts (*Conti et al., 2004*; *Ma et al., 2010*; *Tullio et al., 1997*), which do not remove the maternally deposited mRNA and proteins of the deleted genes. This often hides the essential functions of genes during preimplantation morphogenesis, as is the case, for example, with the cell-cell adhesion molecule CDH1 (*Stephenson et al., 2010*). Moreover, NMHCs could have redundant functions, and gene deletions may trigger compensation mechanisms, which would obscure the function of essential genes (*Rossi et al., 2015*).

In this study, we generated maternal-zygotic deletions of single or double NMHC genes to investigate the molecular control of contractile forces during preimplantation development. We used nested time-lapse microscopy to quantitatively assess the effect of maternal-zygotic deletions at different timescales. This reveals the dominant role of MYH9 over MYH10 in generating the contractility that shapes the mouse blastocyst. In addition, double maternal-zygotic *Myh9* and *Myh10* knockout reveals compensatory mechanisms provided by MYH10 in generating enough contractility for cytokinesis when MYH9 is absent. Moreover, the maternal-zygotic knockout of both *Myh9* and *Myh10* can cause embryos to fail all five successive cleavages, resulting in syncytial single-celled embryos. These single-celled embryos nevertheless initiate lineage specification and blastocoel formation by accumulating fluid into intracellular vacuoles. Therefore, double maternal-zygotic NMHC mutants reveal that fluid accumulation in the blastocyst is a cell-autonomous process. Finally, we confirm this surprising finding by fusing all blastomeres of wild-type (WT) embryos, thereby forming single-celled embryos, which accumulate fluid into inflating vacuoles.

## Results

### NMHC paralogs during preimplantation development

As in humans, the mouse genome contains three genes encoding NMHCs: *Myh9*, *Myh10,* and *Myh14*. To decipher the specific contributions of NMHC paralogs to preimplantation development, we first investigated the expression of *Myh9*, *Myh10,* and *Myh14*.

We performed real-time quantitative PCR (qPCR) at four different stages in order to cover the levels of transcripts at key steps of preimplantation development. We detected high levels of *Myh9* mRNA throughout preimplantation development (*Figure 1A*). Importantly, transcripts of *Myh9* are by far the most abundant among NMHCs at the zygote stage (E0.5), which suggests that *Myh9* is the main NMHC paralog provided maternally. *Myh10* mRNA is detected at very low levels in zygotes before it reaches comparable levels to *Myh9* mRNA at the morula stage (*Figure 1A*). We found that *Myh14* is not expressed during preimplantation stages (*Figure 1A*). Since *Myh14* homozygous mutant mice are viable and show no apparent defects (*Ma et al., 2010*), *Myh14* is unlikely to play an

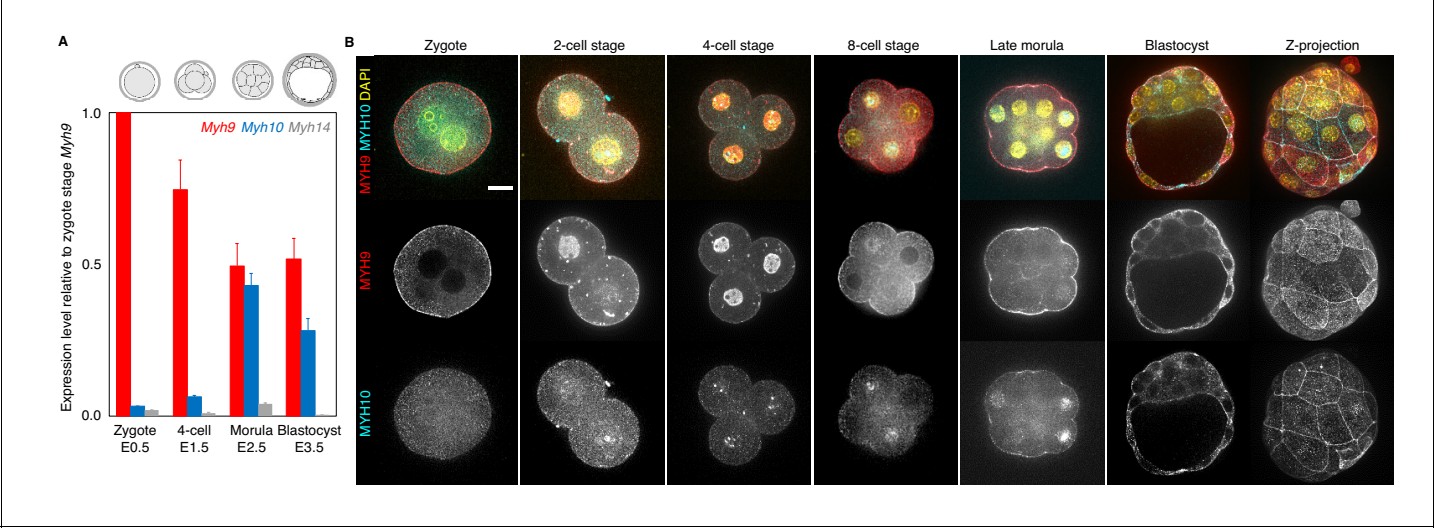

**Figure 1.** Expression of non-muscle myosin II heavy chain (NMHC) paralogs during preimplantation development. (**A**) RT-qPCR of *Myh9* (red), *Myh10* (blue), and *Myh14* (grey) at the zygote (E0.5, n = 234), four-cell (E1.5, n = 159), morula (E2.5, n = 189), and blastocyst (E3.5, n = 152) stages from six independent experiments. Gene expression is normalized to *Gapdh* and shown as the mean ± SEM fold change relative to *Myh9* at the zygote stage. (**B**) Representative images of immunostaining of NMHC paralogs MYH9 (red) and MYH10 (cyan) throughout preimplantation development. DAPI in yellow. Scale bar, 20 μm.

The online version of this article includes the following figure supplement(s) for figure 1:

**Figure supplement 1.** Analysis of the expression of non-muscle myosin II heavy chain (NMHC) paralogs during preimplantation development based on single cell RNA sequencing and MYH9-GFP fluorescence.

important role in preimplantation development. These qPCR measurements show similar trends to available mouse single cell RNA sequencing (scRNA-seq) data (*Figure 1—figure supplement 1A*; *Deng et al., 2014*). In humans, scRNA-seq data indicate similar expression levels between *MYH9* and *MYH10* during preimplantation development and, as in mice, the absence of *MYH14* expression (*Figure 1—figure supplement 1B*; *Yan et al., 2013*). This points to a conserved regulation of NMHC paralogs in mouse and human preimplantation development.

At the protein level, immunostaining of MYH9 becomes visible at the cortex of blastomeres from the zygote stage onwards (*Figure 1B*). On the other hand, MYH10 becomes detectable at the earliest at the 16-cell stage (*Figure 1B*). Finally, we used transgenic mice expressing endogenously tagged MYH9-GFP to assess the relative parental contributions of MYH9 protein (*Figure 1—figure supplement 1C–D*). Embryos coming from MYH9-GFP females show highest levels of fluorescence at the zygote stage, consistent with our qPCR measurement and scRNA-seq data (*Figure 1A*, *Figure 1—figure supplement 1A*). MYH9-GFP produced from the paternal allele is detected at the 4-cell stage and increases until blastocyst stage, reaching levels comparable to blastocyst coming from MYH9-GFP females (*Figure 1—figure supplement 1C–D*).

Together, we conclude that MYH9 and MYH10 are the most abundant NMHCs during mouse and human preimplantation development and that MYH9 is heavily maternally provided at both the transcript and protein level in the mouse embryo (*Figure 1*, *Figure 1—figure supplement 1*).

## Preimplantation development of single maternal-zygotic *Myh9* or *Myh10* mutant embryos

Initial studies of *Myh9* or *Myh10* zygotic knockouts have reported that single zygotic knockouts are able to implant, suggesting they are able to form a functional blastocyst (*Conti et al., 2004*; *Tullio et al., 1997*). A potential lack of phenotype could be due to maternally provided products, which are most abundant in the case of *Myh9* (*Figure 1*, *Figure 1—figure supplement 1*). To eliminate this contribution, we used $Zp3^{Cre/+}$ mediated maternal deletion of conditional knockout alleles of *Myh9* and *Myh10*. This generated either maternal-zygotic (mz) homozygous or maternal only (m) heterozygous knockout embryos for either *Myh9* or *Myh10*. Embryos were recovered at E1.5, imaged throughout the rest of their preimplantation development, stained, and genotyped once WT embryos reached the blastocyst stage. We implemented a nested time-lapse protocol to image each embryo at the long (every 30 min for about 50 hr) and short (every 5 s for 10 min twice for each embryo around the time of their 8-cell stage) timescales. This imaging protocol allowed us to visualize the effect of actomyosin contractility at multiple timescales, as it is involved in pulsatile contractions, cytokinesis, and morphogenesis, which take place on timescales of tens of seconds, minutes, and hours, respectively (*Maître et al., 2015*; *Maître et al., 2016*).

The first visible phenotype concerns the zona pellucida (ZP), which encapsulates the preimplantation embryo. WT and maternal *Myh10* mutants have a spherical ZP, whereas maternal *Myh9* mutants show an irregularly shaped ZP (*Figure 2—figure supplement 1A,B*). This suggests a previously unreported role of contractility in the formation of the ZP during oogenesis that is mediated specifically by maternal MYH9. The abnormally shaped ZP could cause long-term deformation of the embryo, as previous studies indicate that the ZP can influence the shape of the embryo at the blastocyst stage (*Kurotaki et al., 2007*; *Motosugi et al., 2005*).

During compaction, angles formed at the surface of contacting cells increase as blastomeres are pulled into closer contact (*Figure 2A*). In WT embryos, contact angles increase from 87 ± 3° to 147 ± 2° during the 8-cell stage between the third and the fourth cleavage (mean ± SEM from the end of the last cleavage of the third wave to the first one of the fourth wave, 23 embryos, *Figure 2A–B*, *Appendix 1—table 1–2*, *Figure 2—video 1*), as measured previously (*Maître et al., 2015*; *Zhu et al., 2017*). In mz*Myh9* embryos, contact angles only grow from 85 ± 2° to 125 ± 4° (mean ± SEM, 15 embryos, *Figure 2A–B*, *Appendix 1—table 1–2*). This reduced ability to compact is in agreement with previous measurements on heterozygous m*Myh9* embryos, which generate a lower surface tension, and therefore cannot efficiently pull cells into closer contact (*Maître et al., 2016*). During the 8-cell stage, mz*Myh10* embryos initially grow their contacts from 87 ± 5° to 121 ± 4°, similarly to mz*Myh9* embryos (mean ± SEM, 11 embryos, *Figure 2A–B*, *Appendix 1—table 1*). However, unlike mz*Myh9* embryos, mz*Myh10* embryos continue to increase their contact size and reach compaction levels identical to WT embryos by the end of the 16-cell stage (148 ± 4°, mean ± SEM, 11 embryos, *Figure 2A–B*, *Appendix 1—table 1*). Importantly, heterozygous m*Myh9*

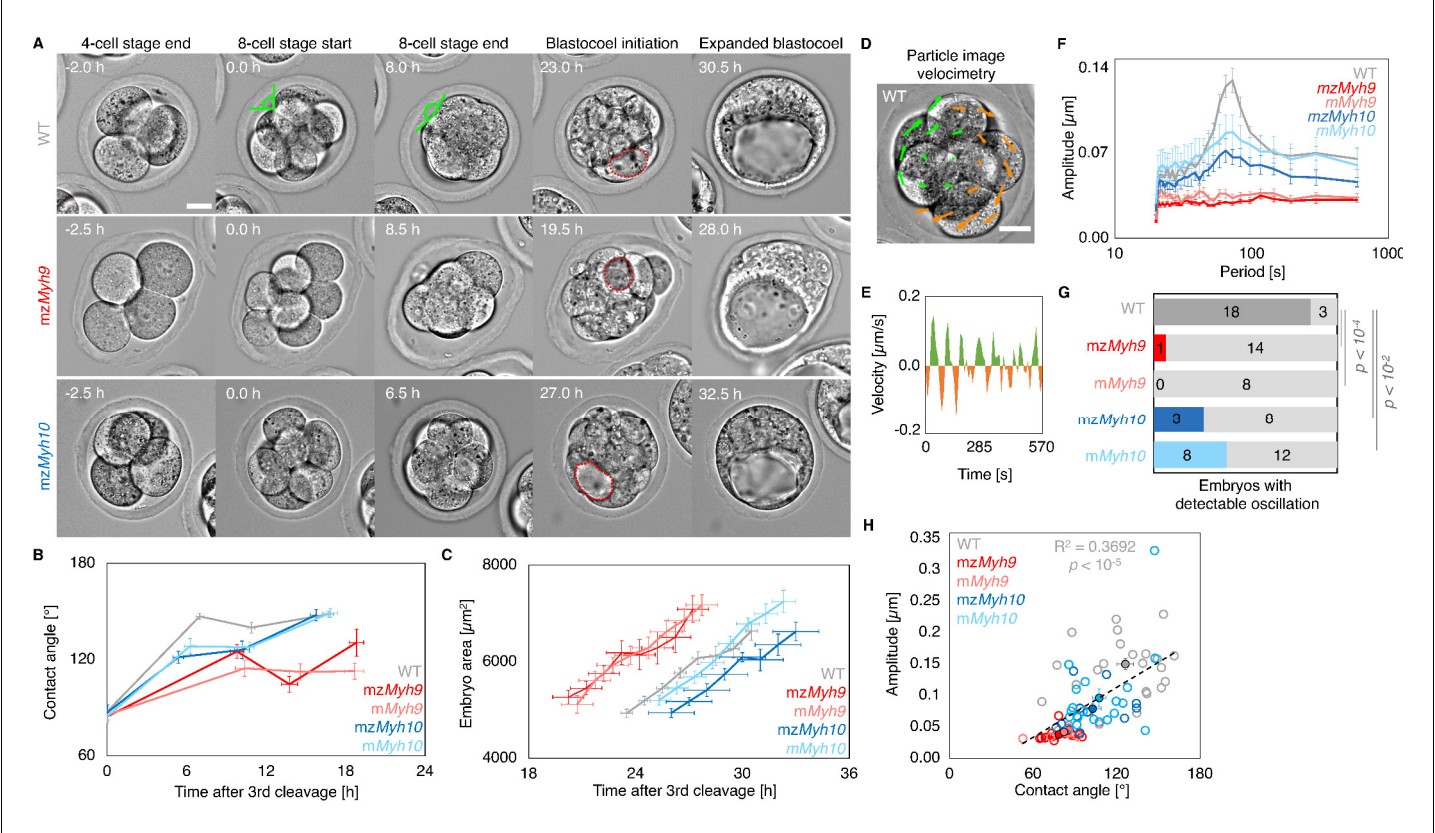

**Figure 2.** Multiscale analysis of morphogenesis in single maternal-zygotic *Myh9* or *Myh10* mutant embryos. (**A**) Representative images of long-term time-lapse of WT, mz*Myh9,* and mz*Myh10* embryos at the end of the 4-cell stage, start and end of the 8-cell stage, at the initiation of blastocoel formation and early blastocyst stage (see also *Figure 2—video 1*). Scale bar, 20 μm. Time in hours after the third cleavage. Green lines mark the contact angles increasing during compaction. Dotted red lines indicate the nascent lumen. (**B**) Contact angle of WT (grey, n = 23, 23, 21, 22), mz*Myh9* (red, n = 15, 15, 10, 8), m*Myh9* (light red, n = 8, 8, 8, 3), mz*Myh10* (blue, n = 11, 11, 11, 11), and m*Myh10* (light blue, n = 20, 20, 20, 20) embryos after the third cleavage, before and after the fourth cleavage and before the fifth cleavage. Data show mean ± SEM. Statistical analyses are provided in *Appendix 1—table 1–2*. (**C**) Embryo growth during lumen formation of WT (grey, n = 20), mz*Myh9* (red, n = 9), m*Myh9* (light red, n = 7), mz*Myh10* (blue, n = 7), and m*Myh10* (light blue, n = 13) embryos measured for seven continuous hours after a lumen of at least 20 μm in diameter is observed. Data show mean ± SEM. (**D**) Representative image of a short-term time-lapse overlaid with a subset of velocity vectors from Particle Image Velocimetry (PIV) analysis. Green for positive and orange for negative Y-directed movement. (**E**) Velocity over time for a representative velocity vector of embryo shown in D and *Figure 2—video 2*. (**F**) Power spectrum resulting from Fourier transform of PIV analysis of WT (grey, n = 21), mz*Myh9* (red, n = 15), m*Myh9* (light red, n = 8), mz*Myh10* (blue, n = 11), and m*Myh10* (light blue, n = 20) embryos. Data show mean ± SEM. (**G**) Proportion of WT (grey, n = 21), mz*Myh9* (red, n = 15), m*Myh9* (light red, n = 8), mz*Myh10* (blue, n = 11), and m*Myh10* (light blue, n = 20) embryos showing detectable oscillations in their power spectrum (see 'Materials and methods'). Chi$^2$ p value comparing to WT is indicated. (**H**) Amplitude of oscillation as a function of the mean contact angle for WT (grey, n = 21), mz*Myh9* (red, n = 15), m*Myh9* (light red, n = 8), mz*Myh10* (blue, n = 11), and m*Myh10* (light blue, n = 20) embryos. Open circles show individual embryos and filled circles give mean ± SEM of a given genotype. Pearson's R$^2$ and p value are indicated. Statistical analyses are provided in *Appendix 1—table 3*.

The online version of this article includes the following video and figure supplement(s) for figure 2:

**Figure supplement 1.** Macroscopic shape analysis of maternal-zygotic *Myh9* and *Myh10* mutant embryos.

**Figure supplement 2.** Morphogenesis of embryos with blastomeres fused at the 4-cell stage.

**Figure 2—video 1.** Preimplantation development of WT, mz*Myh9,* and mz*Myh10* embryos.

https://elifesciences.org/articles/68536#fig2video1

**Figure 2—video 2.** Periodic waves of contraction in WT, mz*Myh9,* mz*Myh10,* and mz*Myh9;*mz*Myh10* embryos.

https://elifesciences.org/articles/68536#fig2video2

**Figure 2—video 3.** Failed cleavage in mz*Myh9* embryos.

https://elifesciences.org/articles/68536#fig2video3

**Figure 2—video 4.** Preimplantation development of control, ¾, and ½ cell number embryos.

https://elifesciences.org/articles/68536#fig2video4

or m*Myh10* embryos show similar phenotypes to their respective homozygous counterparts (*Figure 2B*, *Appendix 1—table 1–2*), suggesting that maternal loss dominates for both NMHC paralogs and that the paternal allele makes little difference. Together, we conclude that maternal MYH9 is essential for embryos to compact fully, whereas MYH10 only regulates the rate of compaction.

During the 8-cell stage, contractility becomes visible on the short timescale as periodic contractions, which we can use to gauge the specific contribution of NMHCs (*Maître et al., 2015*; *Maître et al., 2016*). We performed particle image velocimetry (PIV) and Fourier analysis to evaluate the period and amplitude of periodic movements (*Figure 2D–F*; *Maître et al., 2015*). While 18/21 WT embryos displayed periodic contractions, these were rarely detected in *Myh9* mutants (1/15 mz*Myh9* and 0/8 m*Myh9* embryos) and occasionally in *Myh10* mutants (3/11 mz*Myh10* and 8/20 m*Myh10* embryos; *Figure 2G*, *Figure 2—video 2*). This suggests that contractility is reduced after maternal loss of *Myh10* and is greatly reduced following maternal loss of *Myh9*. This hierarchy in the phenotypes of the NMHC paralog mutants parallels the one observed on the long timescale during compaction (*Figure 2B*). In fact, we found that the amplitude of periodic contractions correlates with the contact angle across the genotypes we considered (*Figure 2H*, 75 embryos, Pearson's R = 0.608, p < 10$^{-5}$, *Appendix 1—table 3*). This analysis across timescales reveals the continuum between the short-term impact of *Myh9* or *Myh10* loss onto contractile movements and the long-term morphogenesis, as previously observed for internalizing ICM cells (*Maître et al., 2016*).

We also noted that the duration of the 8-cell stage is longer in embryos lacking maternal *Myh9* (9.8 ± 0.5 hr from the third to the fourth wave of cleavages, mean ± SEM, 15 embryos) as compared to WT (7.0 ± 0.3 hr, mean ± SEM, 23 embryos; *Figure 2A,B*, *Appendix 1—tables 1–2*, *Figure 2—video 1*). This is not the case for the duration of the fourth wave of cleavages or the ensuing 16-cell stage, which are similar in these different genotypes (*Appendix 1—table 2*). On the long timescale, longer cell cycles could affect the number of cells in *Myh9* mutants. Indeed, when reaching the blastocyst stage, mz*Myh9* embryos count less than half the number of cells than WT (58.1 ± 2.9 cells in 23 WT embryos, as compared to 25.2 ± 2.8 cells in 15 mz*Myh9* embryos, Mean ± SEM; *Figure 3B*). In the time-lapse movies, we did not observe cell death, which could, in principle, also explain a reduced cell number at the blastocyst stage. On the other hand, we did observe reverting cleavages in maternal *Myh9* mutants (*Figure 2—video 3*), effectively reducing cell number at the blastocyst stage. Loss of MYH9 is likely to cause difficulties during cytokinesis (*Taneja et al., 2020*; *Yamamoto et al., 2019*), which could in turn impact cell cycle progression (*Figure 2B*) and explain the significant reduction in cell number in mz*Myh9* embryos. As for mz*Myh10* embryos, we do not observe any cell cycle delay and count blastocysts with the correct cell number (*Figure 2B* and *Figure 3B*), indicating that MYH9 is the primary NMHC powering cytokinesis during mouse preimplantation development.

The second morphogenetic movement consists in the positioning of cells on the inside of the embryo after the fourth cleavage division to form the first lineages of the mammalian embryo. To see if this process is affected in NMHC mutants, we counted the number of inner and outer cells on immunostaining at the blastocyst stage (*Figure 3A–C*). Despite showing less than half the expected cell number, mz*Myh9* blastocysts show the same proportion of inner and outer cells as WT and mz*Myh10* embryos (*Figure 3C*). This suggests that the remaining contractility is sufficient to drive cell internalization or that oriented cell divisions can mitigate the loss of contractility-mediated internalization (*Korotkevich et al., 2017*; *Maître et al., 2016*; *Niwayama et al., 2019*). Outer and inner cells differentiate into TE and ICM, respectively. To assess whether differentiation is affected in NMHC mutants, we performed immunostaining of TE marker CDX2 and ICM marker SOX2 at the blastocyst stage (*Figure 3A*; *Avilion et al., 2003*; *Strumpf et al., 2005*). mz*Myh10* embryos display negligible reduction in CDX2 and SOX2 levels compared to WT embryos (*Figure 3E–G*). On the other hand, we found that mz*Myh9* embryos show lower levels of CDX2 in their outer cells, as measured here and previously for m*Myh9* embryos at the 16-cell stage (*Figure 3F*; *Maître et al., 2016*), and of SOX2 for inner cells compared to WT embryos (*Figure 3G*). This is caused in part by the presence of individual unspecified cells localized both inside and at the surface of mz*Myh9* embryos (*Figure 3D*). To assess whether delayed cell cycle progression may explain the reduced differentiation, we calculated the correlation between the duration of the 8-cell stage and the levels of CDX2 in the TE and of SOX2 in the ICM. This correlation is poor for all maternal *Myh9* mutants (Pearson's

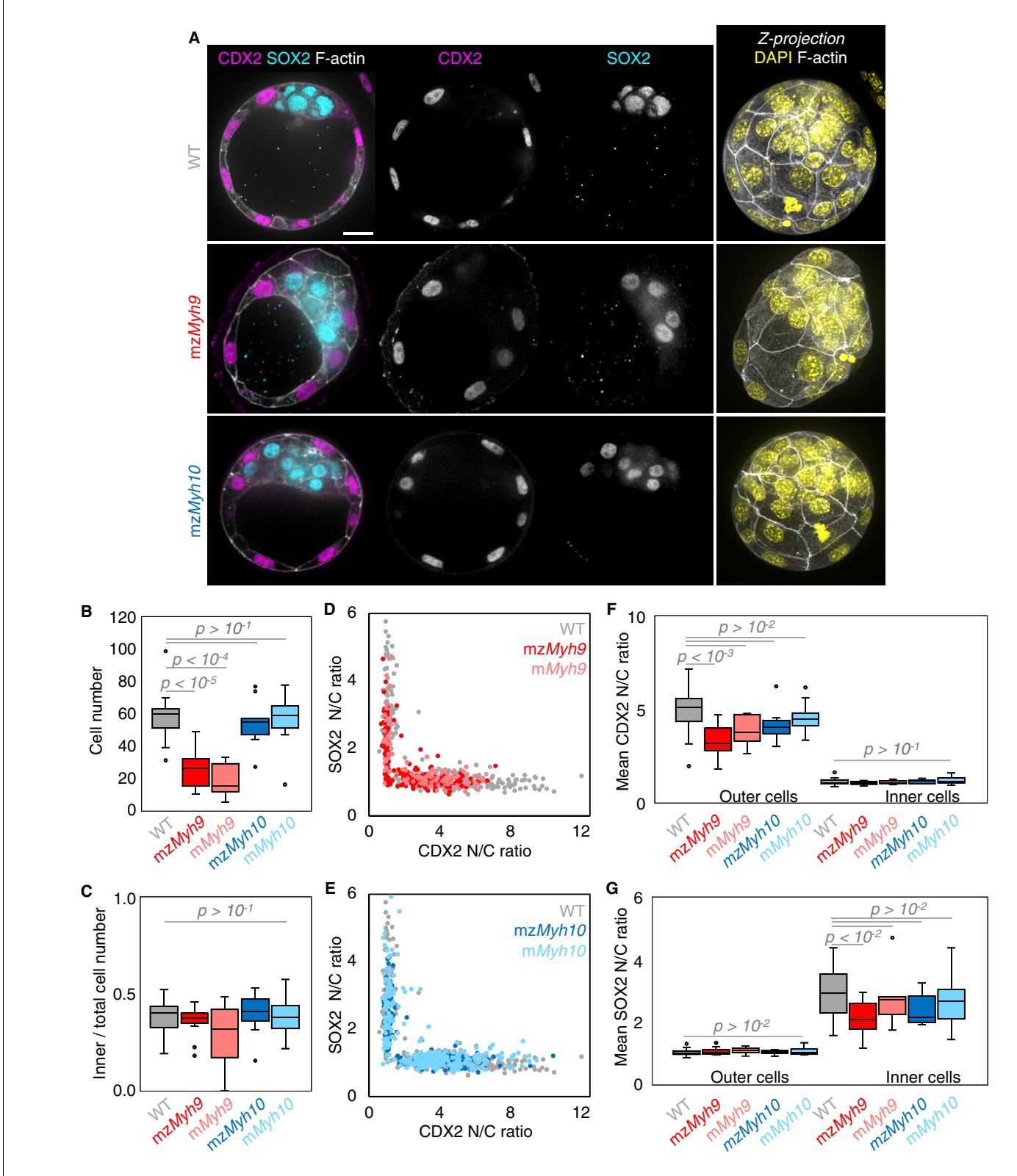

**Figure 3.** Analysis of TE and ICM lineages in single maternal-zygotic *Myh9* or *Myh10* mutant embryos. (**A**) Representative images of WT, mz*Myh9*, and mz*Myh10* embryos stained for TE and ICM markers CDX2 (magenta) and SOX2 (cyan), DAPI (yellow), and F-actin (grey). The same mutant embryos as in *Figure 2A* are shown. Scale bar, 20 μm. (**B-C**) Total cell number (**B**) and proportion of inner cells (**C**) in WT (grey, n = 23), mz*Myh9* (red, n = 15), m*Myh9* (light red, n = 8), mz*Myh10* (blue, n = 11), and m*Myh10* (light blue, n = 19) embryos. (**D-E**) Nuclear to cytoplasmic (N/C) ratio of CDX2 and SOX2

*Figure 3 continued on next page*

*Figure 3 continued*

staining for individual cells from WT (grey, n = 345), mz*Myh9* (red, n = 204), m*Myh9* (light red, n = 95), mz*Myh10* (blue, n = 160), and m*Myh10* (light blue, n = 300) embryos. (F-G) Average N/C ratio of CDX2 (F) and SOX2 (G) staining for outer (left) or inner (right) cells from WT (grey, n = 23), mz*Myh9* (red, n = 15), m*Myh9* (light red, n = 8), mz*Myh10* (blue, n = 11), and m*Myh10* (light blue, n = 19) embryos. Mann-Whitney *U* test p values compared to WT are indicated.

The online version of this article includes the following figure supplement(s) for figure 3:

**Figure supplement 1.** Analysis of YAP localization in maternal-zygotic *Myh9* and *Myh10* mutant embryos.

**Figure supplement 2.** Lineage specification of embryos with blastomeres fused at the 4-cell stage.

R = −0.193 for CDX2 and −0.076 for SOX2, 22 embryos, p>$10^{-2}$), which does not support reduced differentiation levels due to cell cycle delay.

To assess whether signalling upstream of CDX2 and SOX2 expression is affected in NMHC mutants, we stained embryos for the co-transcriptional activator YAP (*Hirate et al., 2013*; *Nishioka et al., 2009*; *Royer et al., 2020*; *Wicklow et al., 2014*). In WT embryos, YAP is enriched specifically in the nuclei of TE cells (*Figure 3—figure supplement 1*; *Hirate et al., 2013*; *Nishioka et al., 2009*; *Wicklow et al., 2014*). Consistent with CDX2 and SOX2 levels, YAP localization is unaffected in mz*Myh10* mutants and reduced in mz*Myh9* mutants (*Figure 3—figure supplement 1*), as observed previously in m*Myh9* mutants at the 16-cell stage (*Maître et al., 2016*). In the mouse preimplantation embryo, the nuclear localization of YAP is promoted by signals from the apical domain (*Hirate et al., 2013*). ML7 treatment, which inhibits the myosin light chain kinase, suggests that contractility may be required for apical domain formation (*Zhu et al., 2017*). However, immunostaining of apico-basal markers reveals that, as in WT, mz*Myh9* and mz*Myh10* embryos accumulate the apical marker PRKCz at the embryo surface (*Figure 4*; *Hirate et al., 2013*). Also, basolateral markers such as the adhesion molecule CDH1 or members of the fluid transport machinery, such as the ATP1A1 subunit of the Na/K pump and aquaporin AQP3, are enriched at basolateral membranes of WT, mz*Myh9*, and mz*Myh10* embryos alike (*Figure 4*; *Barcroft et al., 2003*; *Barcroft et al., 2004*; *Dumortier et al., 2019*; *Stephenson et al., 2010*). Together, these findings suggest that MYH9-mediated contractility is required for the correct expression of lineage markers downstream of YAP but dispensable for the establishment of apico-basal polarity.

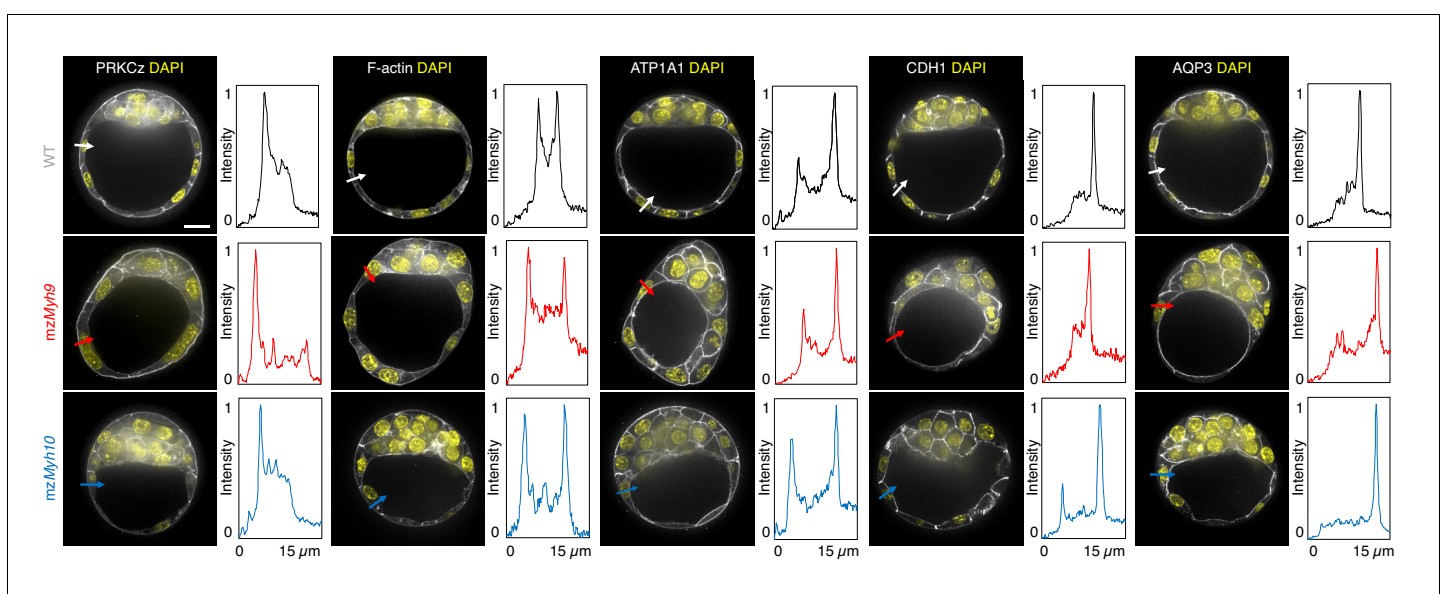

**Figure 4.** Apico-basal polarity of single maternal-zygotic *Myh9* or *Myh10* mutant embryos. Representative images of WT, mz*Myh9*, *and* mz*Myh10* embryos stained for apico-basal polarity markers. From left to right: PRKCz, F-actin, ATP1A1, CDH1, and AQP3 shown in grey. Nuclei stained with DAPI are shown in yellow. Intensity profiles of a representative 15 μm line drawn from the embryo surface towards the interior are shown next to each marker and genotype (arrows). Intensities are normalized to each minimal and maximal signals. Scale bar, 20 μm.

Apico-basal polarity is essential for the final morphogenetic step shaping the blastocyst: the formation of the first mammalian lumen. After the fifth cleavage, the growth of the blastocoel inflates and increases the projected area of WT embryos steadily at $3.9 \pm 0.5$ μm$^2$/min (mean $\pm$ SEM, 20 embryos; *Figure 2C*, *Figure 2—video 1*). During lumen formation, mz*Myh9* and mz*Myh10* embryos inflate at $3.9 \pm 1.0$ and $4.0 \pm 0.5$ μm$^2$/min, respectively, which is similar to WT embryos (mean $\pm$ SEM, 9 and 7 embryos; Student's *t* test compared to WT p>10$^{-2}$, *Figure 2C*). This suggests that blastocoel expansion rate is unaffected in NMHC mutants. Importantly, we observe that the time of lumen opening in mz*Myh9* embryos is not delayed compared to WT embryos (*Figure 2A–C*). In fact, relative to the time of third cleavage, the lumen inflates on average at an earlier stage in mz*Myh9* embryos than in WT (*Figure 2C*). Some mz*Myh9* embryos begin inflating their lumen before the fifth wave of cleavages (5/11; *Figure 2—video 1*), while all WT and mz*Myh10* embryos begin lumen formation after the first cleavages of the fifth wave. This argues against mz*Myh9* embryos having a global delay in their development, despite their longer cell cycles, lower cell number (*Figure 2B* and *Figure 3B*), and impaired differentiation (*Figure 3D,F–G*). Finally, when the lumen becomes sufficiently large, embryos come into contact with the ZP. 3D segmentation of blastocysts reveals that embryos with mutated maternal *Myh9* alleles become less spherical than those with a WT allele (*Figure 2—figure supplement 1C–F*). In fact, the shape of the ZP of mutant embryos at the 2-cell stage correlates with the shape of embryos at the blastocyst stage (*Figure 2—figure supplement 1G*; 67 embryos from all genotypes combined but WT, Pearson's R = −0.739, p<10$^{-2}$), which suggests that the misshapen ZP could deform the embryo. This is consistent with previous studies reporting on *Myh9* mutant embryos being more deformable than WT (*Maître et al., 2016*) and on the influence of the ZP on the shape of the blastocyst (*Kurotaki et al., 2007*; *Motosugi et al., 2005*). Experimental deformation of the embryo was reported to affect cell fate (*Niwayama et al., 2019*; *Royer et al., 2020*). In the range of deformation of the ZP observed in mutant embryos, we found weak correlations between the shape of the ZP and CDX2 or SOX2 levels (*Figure 2—figure supplement 1H–I*, CDX2 in outer cells of 77 embryos from all genotypes combined but WT, Pearson's R = 0.315, p<10$^{-2}$, SOX2 in the inner cells of 58 embryos, Pearson's R = 0.223, p > 10$^{-1}$). This suggests that defects in lineage specification of maternal *Myh9* mutant embryos do not simply result from the deformation of these blastocysts by their ZP. From this multiscale analysis, we conclude that maternal-zygotic loss of *Myh9* or *Myh10* has little impact on lumen formation and that maternal loss of *Myh9* impacts the shape of the blastocyst indirectly by its initial effect on the structure of the ZP.

Maternal *Myh9* mutants show defective cytokinesis and reduced cell number, which could constitute the cause for the defects in morphogenesis and lineage specification. To separate the effects of impaired contractility and defective cell divisions, we decided to reduce cell number by fusing blastomeres of WT embryos together. Reducing cell number by ¼ or ½ at the 4-cell stage results in embryos compacting with the same magnitude and forming a lumen concomitantly to control embryos (*Figure 2—figure supplement 2A–C*, *Figure 2—video 4*, *Figure 3—figure supplement 2B*). Also, embryos with reduced cell numbers differentiate into TE and ICM identically to control embryos as far as lineage proportions and fate marker levels are concerned (*Figure 3—figure supplement 2A,C–E*). Therefore, the morphogenesis and lineage specification defects observed in maternal Myh9 mutants are not likely to result simply from cytokinesis defects and their impact on cell number.

Together, these analyses reveal the critical role of MYH9 in setting global cell contractility on short and long timescales in order to effectively drive compaction, cytokinesis, and lineage specification. Comparably, the function of MYH10 during early blastocyst morphogenesis is less prominent. Importantly, the similarity of maternal homozygous and maternal heterozygous mutants indicates that the preimplantation embryo primarily relies on the maternal pools of MYH9. We conclude that maternally provided MYH9 is the main NMHC powering actomyosin contractility during early preimplantation development.

## Preimplantation development of double maternal-zygotic *Myh9;Myh10* mutants

Despite MYH9 being the main NMHC provided maternally, the successful cleavages, shape changes, and differentiation observed in m*Myh9* and mz*Myh9* embryos suggest that some compensation by MYH10 could occur in these embryos. To test for compensations between the two NMHC paralogs

expressed during preimplantation development, we generated double maternal-zygotic *Myh9* and *Myh10* knockout (mz*Myh9*;mz*Myh10*) embryos.

Nested time-lapse revealed that mz*Myh9*;mz*Myh10* embryos fail most attempts of cytokinesis (*Figure 5A*, *Figure 5—video 1*). In addition to failed cytokinesis, mz*Myh9*;mz*Myh10* embryos show increased cell cycle durations, which are more severe than those for mz*Myh9* embryos (*Figure 5A–B*, *Appendix 1—table 1–2*). This results in mz*Myh9*;mz*Myh10* embryos with only 2.9 ± 0.5 cells when reaching the blastocyst stage (*Figure 6B*). In fact, they occasionally develop into blastocyst-stage embryos consisting of only one single cell (1/8 embryos). Of those which succeed in dividing at least once, compaction after the third cleavage is weak with contact angles for mz*Myh9*;mz*Myh10* embryos capping at 117 ± 4° compared to 147 ± 2° for WT (mean ± SEM, 3 and 23 embryos; *Figure 5B*, *Appendix 1—table 1*, *Figure 5—video 1*). Consistent with previous observations, periodic contractions are undetectable in any of the double mutants that we analyzed (*Figure 5D–E*, *Figure 2—video 2*). Together, this indicates that double maternal-zygotic *Myh9* and *Myh10* mutants generate extremely weak contractile forces compared to WT and single maternal-zygotic *Myh9* or *Myh10* mutants.

When investigating fate specification (*Figure 6*), CDX2 is present in 5/7 embryos as compared to all the 23 WT embryos we analyzed (*Figure 6F*). SOX2 is only detected when embryos succeed in internalizing at least one cell (*Figure 6A,C–E and G*). No maternal *Myh9*;*Myh10* mutant embryo consisting of only two cells contained an inner cell, half of embryos with 3–5 cells contained a single inner cell, and all embryos with six cells or more contained inner cells (*Figure 6D*). Moreover, the levels of CDX2 in outer cells are reduced compared to WT embryos, as observed previously for single maternal-zygotic *Myh9* mutants. Consistently, YAP is enriched in the nucleus of outer cells, albeit at lower levels than in WT (*Figure 3—figure supplement 1*). As for single mutants, the apical marker PRKCz, F-actin, the cell-cell adhesion molecule CDH1, and members of the fluid-pumping machinery ATP1A1 and AQP3 are correctly apico-basally polarized in double mutants (*Figure 7*). The presence of CDX2, nuclear YAP, and the correct organization of apico-basal polarity suggest that mz*Myh9*;mz*Myh10* embryos can initiate differentiation into TE, the epithelium responsible for making the blastocoel. The ability of mz*Myh9*;mz*Myh10* embryos to create de novo a functional polarized epithelium is further supported by the fact that they can eventually proceed with polarized fluid accumulation and create a blastocoel (*Figure 5A*, *Figure 5—video 1*). As observed in single maternal *Myh9* mutants, mz*Myh9*;mz*Myh10* embryos are not delayed to initiate lumen formation despite their extended cell cycles (*Figure 5C*, *Figure 5—video 1*). Also, we measure that mutant embryos grow at rates that are similar to WT ones (5.8 ± 1.5 and 3.9 ± 0.5 $\mu m^2$/min, mean ± SEM, 5 and 20 embryos, Student's *t* test p>$10^{-2}$; *Figure 5C*). This indicates that contractility does not influence the initial growth rate of the blastocoel and that blastocoel formation does not require powerful actomyosin contractility.

Finally, we measure similar metrics for all single and double heterozygous *Myh9* and *Myh10* mutant embryos (*Figure 5*, *Figure 6*, *Appendix 1—table 1*), indicating once again that maternal contributions predominantly set the contractility for preimplantation development. We conclude that after the maternal loss of both *Myh9* and *Myh10*, contractility is almost entirely abolished. This effect is much stronger than that after the loss of maternal MYH9 only and suggests that, in single maternal *Myh9* mutants, MYH10 can compensate significantly, which we would not anticipate from our previous single mz*Myh10* knockout analysis. Therefore, despite MYH9 being the main engine of preimplantation actomyosin contractility, MYH10 ensures a substantial part of blastocyst morphogenesis in the absence of MYH9. The nature and extent of this compensation will need further characterization.

## Preimplantation development of single-celled embryos

Embryos without maternal MYH9 and MYH10 can fail all successive cleavages when reaching the time of the blastocyst stage (18/53 embryos, all *Myh9* and *Myh10* mutant genotypes combined across seven experiments; *Figure 8A*, *Figure 8—video 1*). In single-celled embryos, multiple cleavage attempts at intervals of 10–20 hr can be observed, suggesting that the cell cycle is still operational in those embryos (*Figure 5—video 1*). This is further supported by the presence of multiple large nuclei in single-celled embryos, indicative of preserved genome replication (*Figure 8C*). Interestingly, these giant nuclei contain CDX2 and YAP, but no SOX2, suggesting that, in addition to the cell cycle, the lineage specification program is still partially operational (*Figure 8C–E*, *Figure 3—*

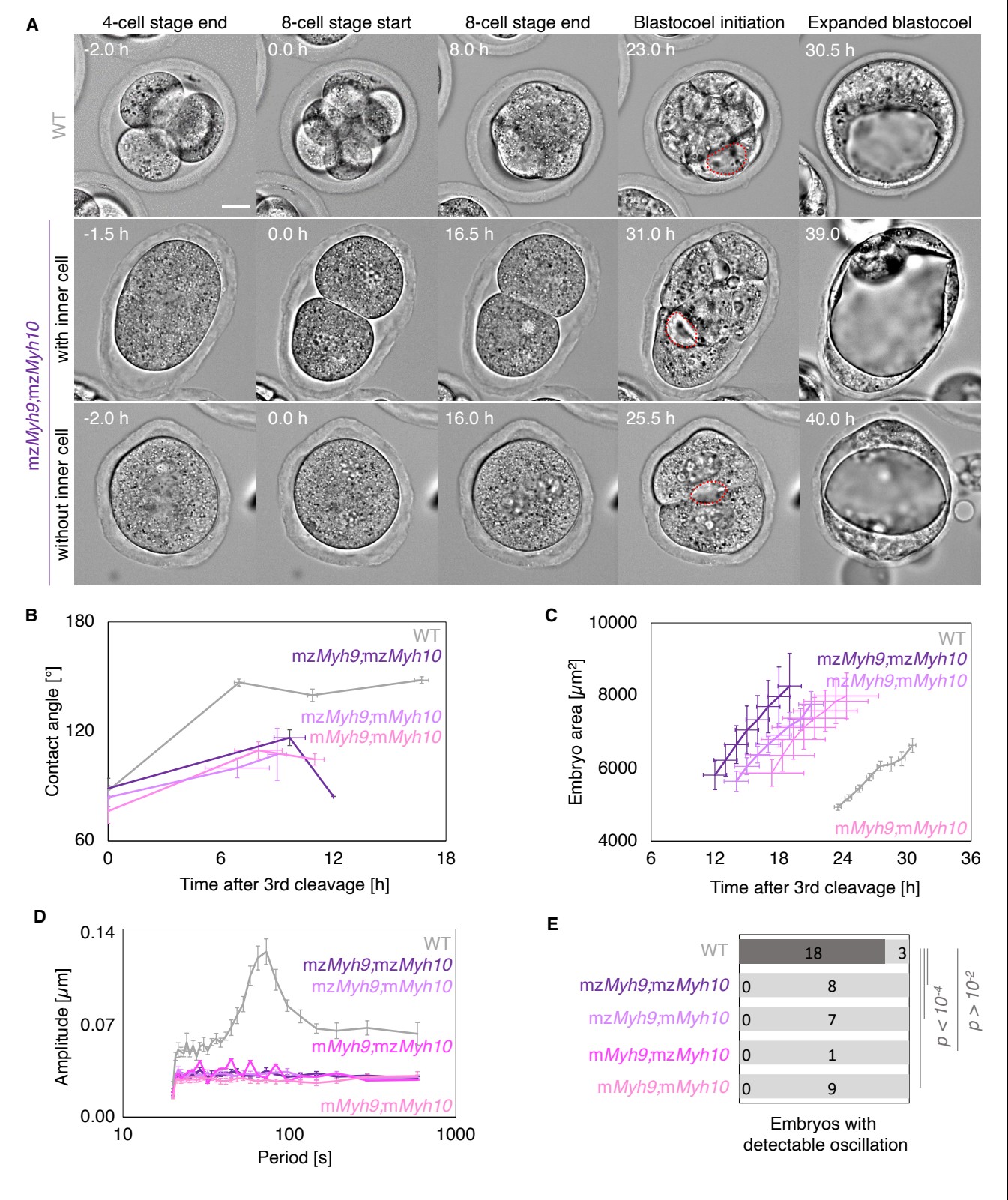

**Figure 5.** Multiscale analysis of morphogenesis in double maternal-zygotic *Myh9* and *Myh10* mutant embryos. (A) Representative images of long-term time-lapse of WT and mz*Myh9*;mz*Myh10* embryos at the end of the 4-cell stage, start and end of the 8-cell stage, at the initiation of blastocoel formation, and early blastocyst stage (*Figure 5—video 1*). The middle row shows an embryo that cleaved once at the time of the third cleavage and twice at the time of the fourth cleavage, which produces an embryo with one inner cell. The bottom row shows an embryo that cleaved once at the

*Figure 5 continued on next page*

*Figure 5 continued*

time of the fourth cleavage, which produces an embryo without inner cell. Scale bar, 20 µm. (B) Contact angle of WT (grey, n = 23, 23, 21, 22), mz*Myh9*;mz*Myh10* (purple, n = 7, 3, 1), mz*Myh9*;m*Myh10* (lilac, n = 7, 4, 2), and m*Myh9*;m*Myh10* (bubblegum, n = 7, 5, 3) embryos after the third cleavage, before and after the fourth cleavage, and before the fifth cleavage, when available. Data show mean ± SEM. Statistical analyses are provided in *Appendix 1—table 1–2*. (C) Embryo growth during lumen formation for WT (grey, n = 20), mz*Myh9*;mz*Myh10* (purple, n = 5), mz*Myh9*;m*Myh10* (lilac, n = 6), and m*Myh9*;m*Myh10* (bubblegum, n = 6) embryos measured for seven continuous hours after a lumen of at least 20 µm is observed. Data show mean ± SEM. (D) Power spectrum resulting from Fourier transform of PIV analysis of WT (grey, n = 21), mz*Myh9*;mz*Myh10* (purple, n = 8), mz*Myh9*;m*Myh10* (lilac, n = 7), m*Myh9*;mz*Myh10* (pink, n = 1), and m*Myh9*;m*Myh10* (bubblegum, n = 9) embryos. Data show mean ± SEM. (E) Proportion of WT (grey, n = 21), mz*Myh9*;mz*Myh10* (purple, n = 8), mz*Myh9*;m*Myh10* (lilac, n = 7), m*Myh9*;mz*Myh10* (pink, n = 1), and m*Myh9*;m*Myh10* (bubblegum, n = 9) embryos showing detectable oscillations in their power spectrum. Chi$^2$ test p value compared to WT is indicated.

The online version of this article includes the following video for figure 5:

**Figure 5—video 1.** Preimplantation development of mz*Myh9*;mz*Myh10* embryos.

https://elifesciences.org/articles/68536#fig5video1

*figure supplement 1*). Strikingly, we also observed that at the time of blastocoel formation, single-celled maternal *Myh9* and *Myh10* mutant embryos begin to swell to sizes comparable to normal blastocysts (*Figure 8A–B*, *Figure 8—video 1*). Single-celled embryos then form, within their cytoplasm, tens of fluid-filled vacuolar compartments, which can inflate into structures comparable in size to the blastocoel (*Figure 8A and C*, *Figure 8—video 1*). The membrane of these cytoplasmic vacuoles contains the fluid pumping machinery ATP1A1 and AQP3, which is specific of basolateral membranes in multicellular embryos of all considered genotypes (*Figure 7*; *Barcroft et al., 2003*; *Barcroft et al., 2004*). This indicates that single-celled embryos are able to establish apico-basal polarity and to direct fluid pumping into a basolateral compartment despite the absence of neighbouring cells. When measuring the growth rates of single-celled embryos, we found that they are comparable to the ones of multiple-celled mutant embryos or WT (4.1 ± 0.8, 5.8 ± 1.5, and 3.9 ± 0.5 µm$^2$/min, mean ± SEM, 4, 5, and 20 embryos, Student's *t* test p > 10$^{-2}$ ; *Figure 8B*), indicating that they are able to accumulate fluid as fast as embryos composed of multiple cells. Therefore, single-celled embryos are able to initiate TE differentiation and display some attributes of epithelial function. Unexpectedly, this also indicates that fluid accumulation during blastocoel formation is cell-autonomous. Together, this further indicates that the developmental program entailing morphogenesis and lineage specification carries on independently of successful cleavages.

To further test whether fluid accumulation is cell-autonomous without disrupting actomyosin contractility, we fused all blastomeres at the late morula stage before lumen formation (*Figure 8—figure supplement 1A*, *Figure 8—video 2*). The nuclei from the fused cells displayed high CDX2 and low SOX2 levels, which indicates that fused embryos retain a TE phenotype (*Figure 8—figure supplement 1C–E*). Moreover, similarly to single-celled embryos resulting from the loss of contractility, single-celled embryos resulting from cell fusion grew in size at rates similar to those of control embryos while forming large fluid-filled vacuoles (3.3 ± 0.4 and 3.7 ± 0.4 µm$^2$/min, 7 and 8 embryos, Student's *t* test p > 10$^{-2}$; *Figure 8—figure supplement 1A–B*, *Figure 8—video 2*). This further confirms that the initiation of fluid accumulation and its rate during blastocoel morphogenesis can rely entirely on transcellular transport and are independent from cell-cell contacts and its associated paracellular transport.

Together, our experiments with single and double maternal-zygotic NMHC knockout embryos reveal that, while MYH9 is the major NMHC powering sufficient actomyosin contractility for blastocyst morphogenesis, MYH10 can significantly compensate the effect of MYH9 loss onto cytokinesis and compaction. Mutant and fused embryos also reveal that even blastocyst-stage embryos made out of a single cell can proceed with the preimplantation program in a timely fashion, as they differentiate into TE and build features of the blastocyst, such as the inflation of fluid-containing compartments. Finally, myosin mutants and fused embryos reveal that the developmental program entailing morphogenesis and lineage specification carries on independently of successful cleavages. This further demonstrates the remarkable regulative capacities of the early mammalian embryo.

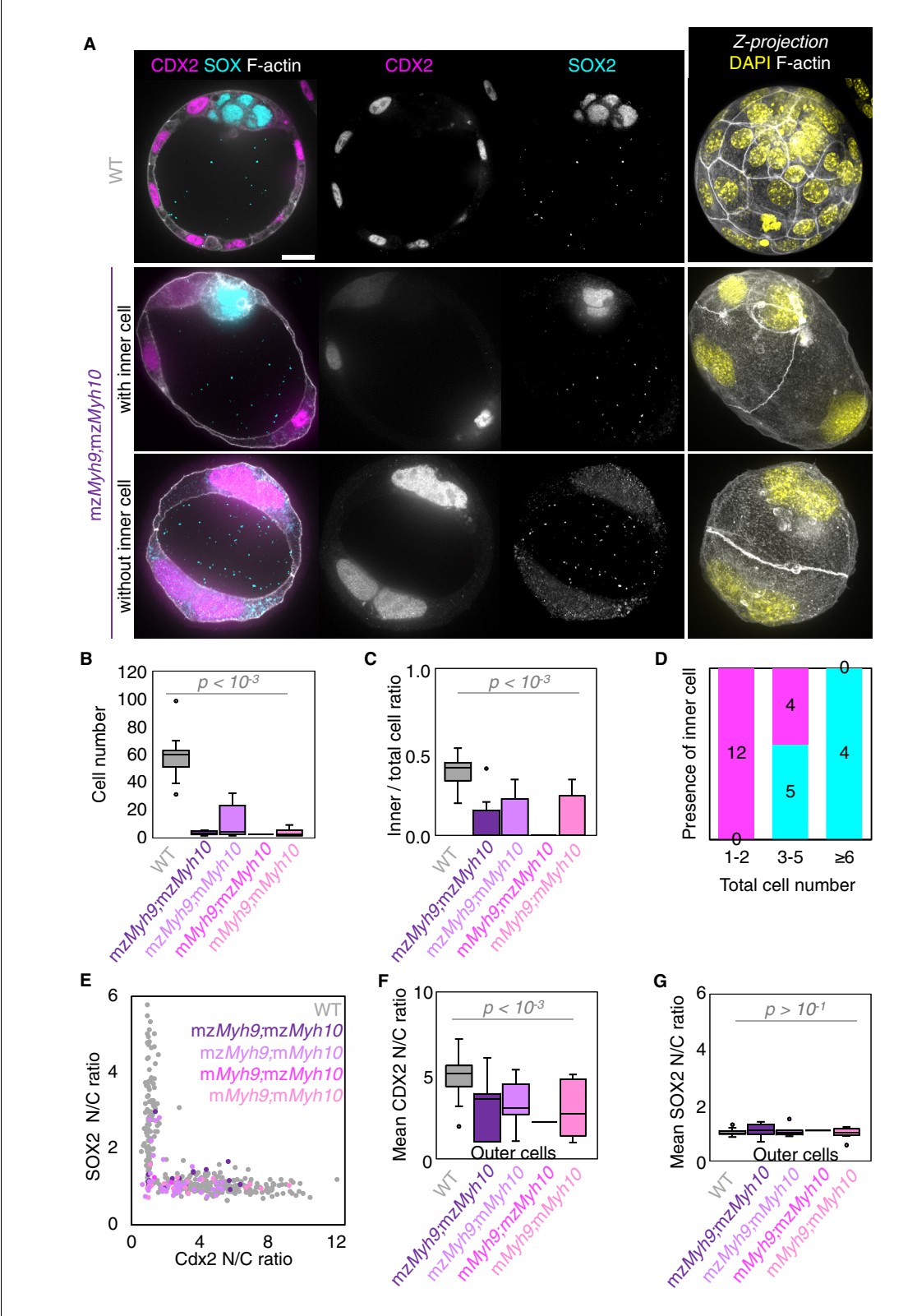

**Figure 6.** Analysis of TE and ICM lineage in double maternal-zygotic *Myh9* and *Myh10* mutant embryos. (**A**) Representative images of WT and mz*Myh9*; mz*Myh10* embryos stained for TE and ICM markers CDX2 (magenta) and SOX2 (cyan), DAPI (yellow), and F-actin (grey). The same mutant embryos as in *Figure 5A* are shown. Scale bar, 20 μm. (**B-C**) Total cell number (**B**) and proportion of inner cells (**C**) in WT (grey, n = 23), mz*Myh9*;mz*Myh10* (purple, n = 8), mz*Myh9*;m*Myh10* (lilac, n = 7), m*Myh9*;mz*Myh10* (pink, n = 1), and m*Myh9*;m*Myh10* (bubblegum, n = 9) embryos. (**D**) Number of maternal *Myh9*; *Figure 6 continued on next page*

*Figure 6 continued*

*Myh10* mutant embryos with (cyan) or without (magenta) inner cells as a function of the total number of cells. (E) Nuclear to cytoplasmic (N/C) ratio of CDX2 and SOX2 staining for individual cells from WT (grey, n = 345), mz*Myh9*;mz*Myh10* (purple, n = 17), mz*Myh9*;m*Myh10* (lilac, n = 41), m*Myh9*; mz*Myh10* (pink, n = 2), and m*Myh9*;m*Myh10* (bubblegum, n = 21) embryos. (F-G) N/C ratio of CDX2 (F) and SOX2 (G) staining for outer cells from averaged WT (grey, n = 23), mz*Myh9*;mz*Myh10* (purple, n = 8), mz*Myh9*;m*Myh10* (lilac, n = 7), m*Myh9*;mz*Myh10* (pink, n = 1), and m*Myh9*;m*Myh10* (bubblegum, n = 9) embryos. Mann-Whitney *U* test p values compared to WT are indicated.

## Discussion

Recent studies have described the critical role of actomyosin contractility in establishing the blastocyst (*Anani et al., 2014*; *Chan et al., 2019*; *Dumortier et al., 2019*; *Maître et al., 2015*; *Maître et al., 2016*; *Samarage et al., 2015*; *Zenker et al., 2018*; *Zhu et al., 2017*). To understand the molecular regulation of this crucial engine of preimplantation development, we have eliminated the maternal and zygotic sources of the NMHCs MYH9 and MYH10 individually and jointly. Maternal-zygotic mutants reveal that actomyosin contractility is primarily mediated by maternal MYH9, which is found most abundantly. Comparatively, loss of *Myh10* has a mild impact on preimplantation development. Nevertheless, double maternal-zygotic mutants reveal that in the absence of MYH9, MYH10 plays a critical role in ensuring sufficient contractility to power cleavage divisions. Furthermore, double maternal-zygotic mutants bring to light the remarkable ability of the preimplantation embryo to carry on with its developmental program even when reduced to a single-celled embryo. Indeed, single-celled embryos, obtained from NMHC mutants or from the fusion of all cells of WT embryos, display evident signs of differentiation and initiate lumen formation by accumulating fluid in a timely fashion. Therefore, NMHC mutants and fused embryos reveal that the developmental program entailing morphogenesis and lineage specification carries on independently of successful cleavages.

Despite the importance of contractility during the formation of the blastocyst, maternal-zygotic mutants of NMHCs are yet to be characterized. The phenotypes we report here further confirm some of the previously proposed roles of contractility during preimplantation development and provide molecular insights on which NMHC paralog powers contractility. We observed reduced compaction when *Myh9* is maternally deleted and slower compaction when *Myh10* is deleted (*Figure 2A–B*). This is in agreement with cell culture studies, in which *Myh9* knockdown reduces

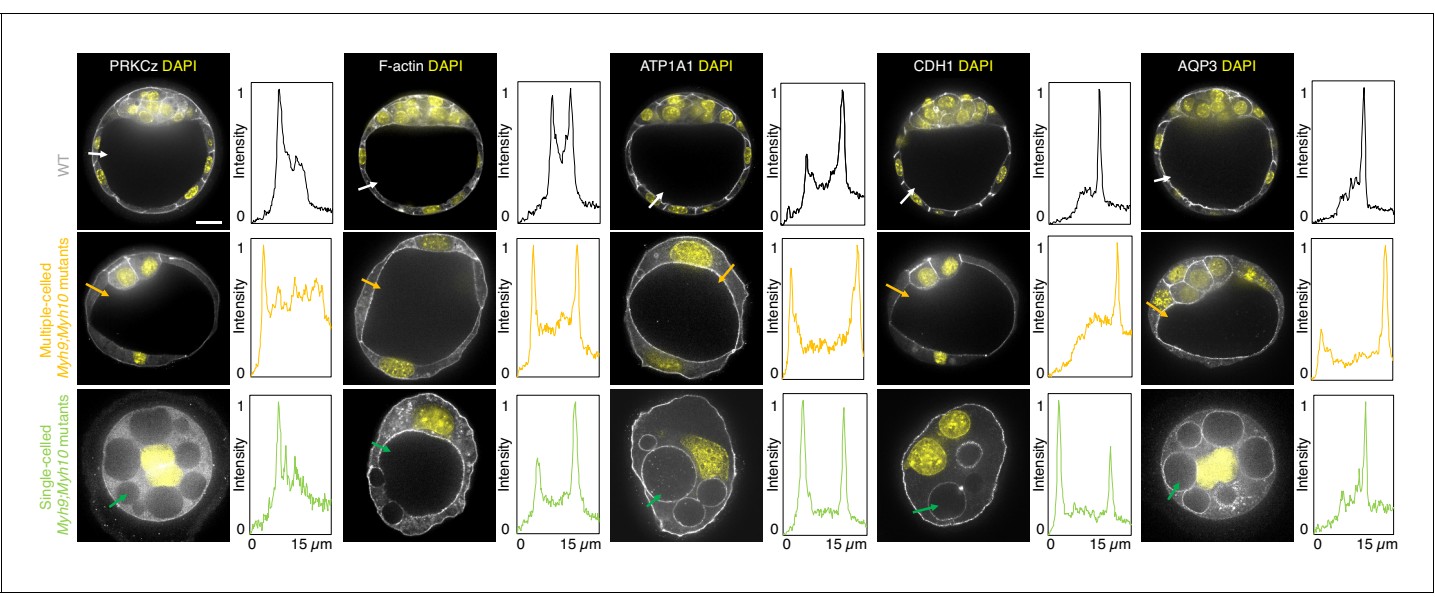

**Figure 7.** Apico-basal polarity of maternal *Myh9*;*Myh10* mutant embryos composed of multiple cells or a single cell. Representative images of WT, multiple- or single-celled *Myh9*;*Myh10* mutant embryos stained for apico-basal polarity markers. From left to right: PRKCz, F-actin, ATP1A1, CDH1, and AQP3 shown in grey. Nuclei stained with DAPI shown in yellow. Intensity profiles of a representative 15 µm line drawn from the embryo surface towards the interior are shown next to each marker and genotype (arrows). Intensities are normalized to each minimal and maximal signals. Scale bar, 20 µm.

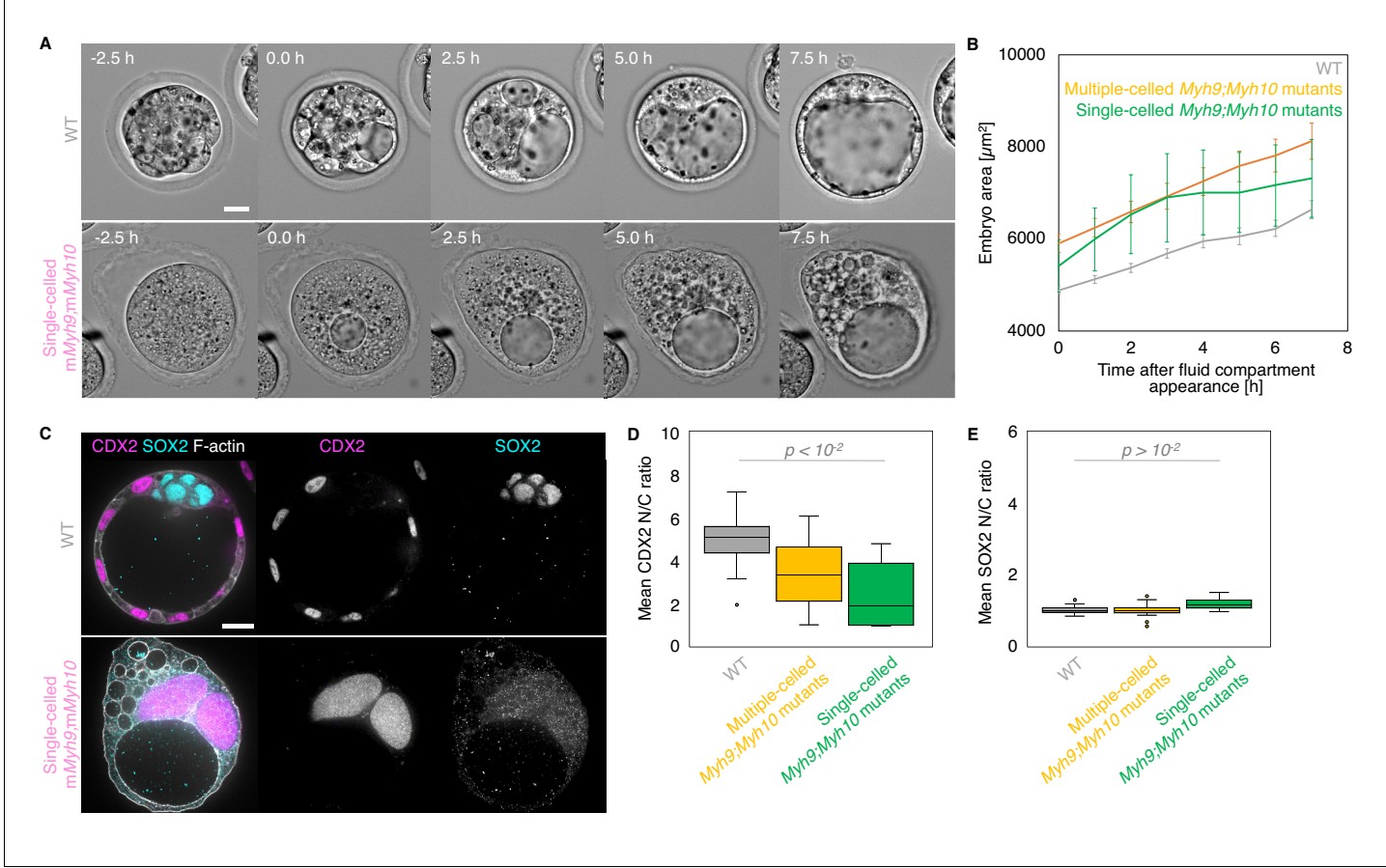

**Figure 8.** Single-celled embryos at the blastocyst stage. (**A**) Representative images of long-term time-lapse of WT and single-celled m*Myh9*;m*Myh10* embryos at the onset of fluid accumulation (*Figure 8—video 1*). Scale bar, 20 μm. (**B**) Embryo growth curves during fluid accumulation for WT (grey, n = 20) and multiple- (yellow, n = 13) or single-celled *Myh9;Myh10* (green, n = 4) mutant embryos measured for seven continuous hours after a lumen of at least 20 μm is observed. Data show mean ± SEM. (**C**) Representative images of WT and single-celled m*Myh9*;m*Myh10* embryos stained for TE and ICM markers CDX2 (magenta) and SOX2 (cyan), DAPI (yellow), and F-actin (grey). The same mutant embryos as in A are shown. Scale bar, 20 μm. (**D-E**) N/C ratio of CDX2 (**D**) and SOX2 (**E**) staining for outer cells from WT (grey, n = 23) and multiple- (yellow, n = 18) or single-celled *Myh9;Myh10* (green, n = 6) embryos. Mann-Whitney *U* test p values compared to WT are indicated.

The online version of this article includes the following video and figure supplement(s) for figure 8:

**Figure supplement 1.** Morphogenesis and lineage specification of embryos with all blastomeres fused at the late morula stage.

**Figure 8—video 1.** Preimplantation development of an m*Myh9*;m*Myh10* embryo failing all successive cleavages.

https://elifesciences.org/articles/68536#fig8video1

**Figure 8—video 2.** Fluid accumulation in control and single-celled fused embryos.

https://elifesciences.org/articles/68536#fig8video2

contact size whereas *Myh10* knockdown does not (*Heuzé et al., 2019*; *Smutny et al., 2010*). This is also in agreement with previous studies using inhibitory drugs on the preimplantation embryo. Compaction is prevented, and even reverted, when embryos are treated with blebbistatin, a drug directly inhibiting NMHC function (*Maître et al., 2015*), or ML7, an inhibitor of the myosin light chain kinase (*Zhu et al., 2017*). However, zygotic injection of siRNA targeting *Myh9* did not affect compaction (*Samarage et al., 2015*; *Zenker et al., 2018*), further indicating that MYH9 acts primarily via its maternal pool. Similarly, siRNA-mediated knockdown of a myosin regulatory light chain *Myl12b* did not affect compaction (*Zhu et al., 2017*). Therefore, for efficient reduction of contractility with molecular specificity, maternal depletion is required.

The phenotypes of the maternal-zygotic NMHC mutants can also appear in contradiction with other conclusions from previous studies. Drugs were used to test the role of contractility in apico-basal polarity establishment with contradictory results. ML7, but not blebbistatin, was reported to

block de novo cell polarization (*Zhu et al., 2017*). This discrepancy might be due to ML7 inhibiting other kinases than MLCK such as PKC (*Bain et al., 2003*), which is required for apical domain formation (*Zhu et al., 2017*). Also, blebbistatin was reported to block the maturation of the apical domain (*Zenker et al., 2018*). The apical domain is essential for TE differentiation and for polarized fluid transport (*Laeno et al., 2013*; *Hirate et al., 2013*). We observed that even the most severely affected NMHC mutant embryos are able to establish and maintain apico-basal polarity, specific expression of TE markers and, most importantly, a functional polarized fluid transport (*Figure 8*). This argues against the requirement of contractility for de novo polarization and apical domain maturation.

Besides compaction defects, reduced cell number is one of the major effects caused by reduced contractility in the maternal-zygotic knockout of NMHCs. Loss of MYH9 alone, but not of MYH10 alone, causes cytokinesis defects (*Figures 2–3*, *Figure 2—videos 1* and *3*). This could simply be explained by the higher levels of maternal MYH9 (*Figure 1*). In vitro studies also noted that MYH9, but not MYH10, is key to power the ingression of the cleavage furrow (*Taneja et al., 2020*). This recent study also noted that MYH10 can compensate the absence of MYH9 with increased cortical recruitment (*Taneja et al., 2020*). Consistently, we observed much more severe division failures when both *Myh9* and *Myh10* are maternally removed (*Figures 5*, *6* and *8*). This reveals significant compensation between NMHCs during preimplantation development. Whether this compensation takes place by a change in subcellular localization or by changes in expression level will require further studies.

Reduced cell number could have a major impact during very early development, when the zygote needs to produce enough cells for developmental patterning and morphogenesis. During normal development, the mouse embryo paces through the successive morphogenetic steps of compaction, internalization, and lumen formation at the rhythm of the third, fourth, and fifth waves of cleavage (*Maître, 2017*; *Płusa and Piliszek, 2020*; *Rossant, 2016*; *White et al., 2018*). Such concomitance may initially suggest that the preimplantation program is tightly linked to cleavages and cell number. However, seminal studies have established that the mammalian embryo is regulative and can build the blastocyst correctly with either supernumerary or fewer cells (*Smith and McLaren, 1977*; *Snow, 1973*; *Tarkowski, 1959*; *Tarkowski, 1961*; *Tarkowski and Wróblewska, 1967*). For example, aggregating multiple embryos results in compaction and lumen formation after the third and fifth cleavage, respectively, rather than when the chimeric embryos are composed of 8 and 32 cells (*Tarkowski, 1961*). Disaggregation leads to the same result with halved or quartered embryos compacting and forming a lumen at the correct embryonic stages rather than when reaching a defined cell number (*Tarkowski, 1959*; *Tarkowski and Wróblewska, 1967*). Contrary to disaggregated embryos, NMHC mutant embryos show reduced cell numbers without affecting the amount of cellular material (*Figure 2—figure supplement 1*). As observed in other species, the pace of development during mouse cleavage stages is thought to depend on the size ratio between the nucleus and the cytoplasm (*Tsichlaki and FitzHarris, 2016*). Evidences of nuclear divisions are visible even in the case of single-celled embryos (*Figure 8—video 1*), which suggests that the nuclear to cytoplasmic ratio may increase during the development of single-celled embryos. In line with these previous studies, the present experiments further confirm that compaction, internalization, and lumen formation take place without the expected cell number. Therefore, although cleavages and morphogenetic events are concomitant, they are not linked. Importantly, affecting blastocyst cell number through maternal *Myh9* loss or fusion of blastomeres at the 4-cell stage preserves the proportion of cells allocated to TE and ICM lineages (*Figure 3C* and *Figure 3—figure supplement 2D*). A recent study proposed that this proportion is controlled by the growth of the lumen (*Chan et al., 2019*). This is compatible with our observation that lumen growth is not affected by the loss of *Myh9* or cell fusion (*Figure 2C* and *Figure 2—figure supplement 2C*). However, lineage proportions break down when both *Myh9* and *Myh10* are maternally lost (*Figure 6C*). With about three cells on an average (*Figure 6C–D*), a majority of double mutants fail to internalize cells. As observed for doublets of 16-cell stage blastomeres, in principle, two cells should be enough to internalize one cell by an entosis-like process (*Maître et al., 2016*; *Overholtzer et al., 2007*). With extremely low contractility, double mutants may fail to raise contractility above the tension asymmetry threshold, enabling internalisation (*Maître et al., 2016*). As for oriented cell division (*Korotkevich et al., 2017*), this mechanism needs at least two cells to push one dividing cell within the cell-cell contact area. Together, this explains the absence of inner cells in mutant embryos with fewer than three cells.

Disaggregation experiments were key to reveal some of the cell-autonomous aspects of blastocyst formation: a single 8-cell stage blastomere can polarize (*Ziomek and Johnson, 1980*) and increase its contractility (*Maître et al., 2015*). Our observations on single-celled embryos reveal that fluid accumulation is also a cell-autonomous process (*Figure 8*). In fact, single-celled embryos inflate at rates similar to control embryos, implying that cell-cell contacts, closed by tight junctions at the embryo surface, are not required for accumulating fluid within blastomeres. Therefore, blastocoel components may be transported exclusively transcellularly, whereas paracellular transport through tight junctions may be negligible.

Whether single-celled embryos resulted from systematically failed cytokinesis or from cell fusion, these embryos grow at a similar rate, which is also the same as control embryos and other embryos composed of a reduced cell number (*Figure 2C*, *Figure 5C*, *Figure 8B*, *Figure 8—figure supplement 1B*, and *Figure 2—figure supplement 2C*). This suggests that fluid accumulation is a robust machinery during preimplantation development. Interfering with aquaporins, ion channels and transporters affect fluid accumulation, but the transcellular and paracellular contributions remain unclear (*Barcroft et al., 2003*; *Kawagishi et al., 2004*; *Schliffka and Maître, 2019*). Future studies will be needed to understand what sets robust transcellular fluid accumulation during preimplantation development. The ability of single cells to polarize and inflate a lumen was recently demonstrated in vitro by plating hepatocytes onto CDH1-coated substrates (*Zhang et al., 2020*). In this study, fluid accumulation in single-celled embryos does not result in the formation of a lumen but in the formation of inflating vacuoles. Vacuoles were also reported to appear occasionally in mutant mouse embryos with defective cell-cell adhesion (*Bedzhov et al., 2012*; *Stephenson et al., 2010*). Similarly, vacuole formation in case of acute loss of cell-cell adhesion has been studied in vitro (*Vega-Salas et al., 1988*). When apical lumens are disrupted, the apical compartment is internalized, and this can result in fluid accumulation within the vacuolar apical compartment (VAC) (*Vega-Salas et al., 1988*). Analogous VACs were proposed to coalesce into the lumen of endothelial tubes upon cell-cell contact in vivo (*Kamei et al., 2006*). The molecular characteristics and dynamics of these structures and their empty counterparts, the so-called apical membrane initiation sites, have been extensively studied in vitro and proven helpful to better understand apical lumen formation (*Bryant et al., 2010*; *Overeem et al., 2015*). In the case of the blastocyst, which forms a lumen on its basolateral side (*Dumortier et al., 2019*; *Schliffka and Maître, 2019*), such vacuoles may constitute an equivalent *vacuolar basolateral compartment*. Similar to apical lumen, further characterization of the molecular identity and dynamic behaviors of these vacuoles may prove useful to better understand fluid accumulation during basolateral lumen formation. Studying transcellular fluid transport and the molecular machinery of basolateral lumen formation will be greatly facilitated by the fusion approach reported in the present study (*Figure 8—figure supplement 1*).

# Materials and methods

## Embryo work
### Recovery and culture
All the animal works were performed in the animal facility at the Institut Curie, with permission from the institutional veterinarian overseeing the operation (APAFIS #11054–2017082914226001). The animal facilities are operated according to international animal welfare rules.

Embryos were isolated from superovulated female mice mated with male mice. Superovulation of female mice was induced by intraperitoneal injection of 5 international units (IU) pregnant mare's serum gonadotropin (Ceva, Syncro-part), followed by intraperitoneal injection of 5 IU human chorionic gonadotropin (MSD Animal Health, Chorulon) 44–48 hr later. Embryos at E0.5 were recovered from plugged females by opening the ampulla followed by a brief treatment with 37°C 0.3 mg/mL hyaluronidase (Sigma, H4272-30MG) and washing in 37°C FHM. Embryos were recovered at E1.5 by flushing oviducts and at E2.5 and E3.5 by flushing oviducts and uteri from plugged females with 37°C FHM (Millipore, MR-122-D) using a modified syringe (Acufirm, 1400 LL 23).

Embryos were handled using an aspirator tube (Sigma, A5177-5EA) equipped with a glass pipette pulled from glass micropipettes (Blaubrand intraMark or Warner Instruments).

Embryos were placed in KSOM (Millipore, MR-107-D) supplemented with 0.1% BSA (Sigma, A3311) in 10 µL droplets covered in mineral oil (Acros Organics). Embryos are cultured in an incubator under a humidified atmosphere supplemented with 5% $CO_2$ at 37°C.

To remove the ZP, embryos were incubated for 45–60 s in pronase (Sigma, P8811).

Blastomeres were fused using the GenomONE-CF FZ SeV-E cell fusion kit (CosmoBio, ISK-CF-001-EX). HVJ envelope was resuspended following manufacturer's instructions and diluted in FHM for use. To fuse the blastomeres of embryos at the 4-cell stage, embryos were incubated in 1:60 HVJ envelope/FHM for 15 min at 37°C followed by washes in KSOM. To fuse blastomeres at the morula stage, embryos were treated in the same manner in 1:50 HVJ envelope/FHM. Fusion typically completes ~30 min after the treatment.

For imaging, embryos were placed in 3.5 or 5 cm glass-bottom dishes (MatTek).

## Mouse lines

Mice used were of 5 weeks old and above.

(C57BL/6xC3H) F1 hybrid strain was used for WT.

To remove LoxP sites specifically in oocytes, $Zp3^{Cre/+}$ (Tg(Zp3-cre)93Knw) mice were used (*de Vries et al., 2000*).

To generate mz*Myh9* embryos, $Myh9^{tm5RSad}$ mice were used (*Jacobelli et al., 2010*) to breed $Myh9^{tm5RSad/tm5RSad}$; $Zp3^{Cre/+}$ females with $Myh9^{+/-}$ ±. To generate mz*Myh10* embryos, $Myh10^{tm7Rsad}$ mice were used (*Ma et al., 2009*) to breed $Myh10^{tm7Rsad/tm7Rsad}$; $Zp3^{Cre/+}$ females with $Myh10^{+/-}$ ±. To generate mz*Myh9*;mz*Myh10* embryos, $Myh9^{tm5RSad/tm5RSad}$; $Myh10^{tm7Rsad/tm7Rsad}$; $Zp3^{Cre/+}$ females were mated with $Myh9^{+/-}$; $Myh10^{+/-}$ males. To generate embryos with a maternal *Myh9-GFP* allele, $Myh9^{tm8.1RSad}$ ($Gt(ROSA)26Sor^{tm4(ACTB-tdTomato,-EGFP)Luo}$) females were mated with WT males; to generate embryos with a paternal *Myh9-GFP* allele, WT females were mated with $Myh9^{tm8.1RSad}$ ($Gt(ROSA)26Sor^{tm4(ACTB-tdTomato,-EGFP)Luo}$) males (*Muzumdar et al., 2007*; *Zhang et al., 2012*). For fusion at the morula stage, ($Gt(ROSA)26Sor^{tm4(ACTB-tdTomato,-EGFP)Luo}$) mice were used (*Muzumdar et al., 2007*).

| Mouse strain | RRID |
|---|---|
| (C57BL/6xC3H) F1 | MGI:5650923 |
| Tg(Zp3-cre)93Knw | MGI:3835429 |
| $Myh9^{tm5RSad/tm5RSad}$ | MGI:4838530 |
| $Myh10^{tm7Rsad/tm7Rsad}$ | MGI:4443040 |
| $Myh9^{tm8.1RSad}$ | MGI:5499741 |
| $Gt(ROSA)26Sor^{tm4(ACTB-tdTomato,-EGFP)Luo}$ | IMSR_JAX:007676 |

## Immunostaining

Embryos were fixed in 2% PFA (Euromedex, 2000-C) for 10 min at 37°C, washed in PBS, and permeabilized in 0.1% (SOX2, CDX2, and YAP primary antibodies) or 0.01% (all other primary antibodies) Triton X-100 (Euromedex, T8787) in PBS (PBT) at room temperature before being placed in blocking solution (PBT with 3% BSA) at 4°C for 2–4 hr. Primary antibodies were applied in blocking solution at 4°C overnight. After washes in PBT at room temperature, embryos were incubated with secondary antibodies, DAPI, and/or phalloidin in blocking solution at room temperature for 1 hr. Embryos were washed in PBT and imaged immediately after.

| Primary antibodies | Dilution | Provider | RRID |
|---|---|---|---|
| CDX2 | 1:200 | Abcam, ab157524 | AB_2721036 |
| SOX2 | 1:100 | Abcam, ab97959 | AB_2341193 |
| YAP | 1:100 | Abnova, H00010413-M01 | AB_535096 |
| Phospho-MYH9 (Ser1943) | 1:200 | Cell Signaling, 5026 | AB_10576567 |

*Continued on next page*

*Continued*

| Primary antibodies | Dilution | Provider | RRID |
|---|---|---|---|
| MYH10 | 1:200 | Santa Cruz, sc-376942 | |
| PRKCz (H-1) | 1:50 | Santa Cruz, sc-17781 | AB_628148 |
| AQP3 | 1:500 | Novusbio, NBP2-33872 | |
| CDH1 | 1:500 | eBioscience,14-3249-82 | |
| ATP1A1 | 1:100 | Abcam, ab76020 | AB_1310695 |

| Secondary antibodies and dyes | Dilution | Provider | RRID |
|---|---|---|---|
| Alexa Fluor Plus 488 anti-mouse | 1:200 | Invitrogen, A32723 | AB_2633275 |
| Alexa Fluor 546 anti-mouse | 1:200 | Invitrogen, A11003 | AB_2534071 |
| Alexa Fluor Plus 488 anti-rabbit | 1:200 | Invitrogen, A32731 | AB_2633280 |
| Alexa Fluor Plus 546 anti-rabbit | 1:200 | Invitrogen, A11010 | AB_2534077 |
| Alexa Fluor 633 anti-rat | 1:200 | Invitrogen, A21094 | AB_141553 |
| Alexa Fluor 488 anti-rat | 1:200 | Invitrogen, A11006 | AB_2534074 |
| Alexa Fluor 633 phalloidin | 1:200 | Invitrogen, A22284 | |
| 4′,6-diamidino-2-phenylindole (DAPI) | 1:1000 | Invitrogen, D1306 | AB_2629482 |

## Single embryo genotyping

DNA extraction was performed on single fixed embryos, in 10 µL of DNA extraction buffer containing 10 mM Tris-HCl (pH 8, Sigma, T2694), 50 mM KCl (Sigma, 60142), 0.01% gelatine (Sigma, G1393), 0.1 mg/mL proteinase K (Sigma, P2308) at 55℃ for 90 min, followed by deactivation of the proteinase K at 90℃ for 10 min. 2 µL of this DNA extract was used in PCR reactions.

To assess the *Myh9* genotype, a preamplification PCR was performed using forward (fw) primer GGGACCCACTTTCCCCATAA/reverse (rev) primer GTTCAACAGCCTAGGATGCG at a final concentration of 0.4 µM. The PCR program is as follows: denaturation at 94℃ 4 min; 35 cycles of 94℃ 1 min, 58℃ 1 min, 72℃ 3:30 min, 72℃ 1 min; final elongation step at 72℃ 7 min. Subsequently, 2 µL of the PCR product was directly used as a template for two independent PCR amplifications to detect either a 592 bp amplicon for the WT allele, with fw primer GGGACACAGTTGAATCCCTT/rev primer ATGGGCAGGTTCTTATAAGG or a 534 bp amplicon for a mutant allele, with fw primer GGGACACAGTTGAATCCCTT/rev primer CATCCTGTGGAGAGTGAGAGCAC at a final concentration of 0.4 µM. PCR program is as follows: denaturation at 94℃ for 4 min; 35 cycles of 94℃ 1 min, 58℃ 2 min, 72℃ 1 min; final elongation step at 72℃ 7 min.

To assess the *Myh10* genotype, a preamplification PCR was performed using fw primer GGCCCCCATGTTACAGATTA/rev primer TTTCCTCAACATCCACCCTCTG at a final concentration of 0.4 µM. The PCR program is as follows: denaturation at 94℃ for 4 min; 35 cycles of 94℃ 1 min, 58℃ 1 min, 72℃ 2 min, 72℃ 1 min; final elongation step at 72℃ 7 min. Subsequently, 2 µL of the PCR product was directly used as a template for a PCR amplification using fw primer 1 TAGCGAAGGTCTAGGGGAATTG/fw primer 2 GACCGCTACTATTCAGGACTTATC/rev primer CAGAGAAACGATGGGAAAGAAAGC at a final concentration of 0.4 µM. PCR program is as follows: denaturation at 94℃ for 4 min; 35 cycles of 94℃ 1 min, 58℃ 1:30 min, 72℃ 1 min; final elongation step at 72℃ 7 min, resulting in a 230 bp amplicon for a WT allele and a 630 bp amplicon for a mutant allele.

## Quantitative RT-PCR

To extract total RNA, embryos were collected in 3 µL of PBS, frozen on ice or snap-frozen on dry ice and stored at −80℃ until further use. Total RNA extraction was performed using the PicoPure RNA Isolation Kit (ThermoFisher Scientific, KIT0204) according to manufacturer's instructions. DNase treatment was performed during the extraction, using RNase-Free DNase Set (QIAGEN, 79254).

cDNA was synthesized with random primers (ThermoFisher Scientific, 48190011), using the SuperScript III Reverse Transcriptase kit (ThermoFisher Scientific, 18080044) on all the extracted RNA, according to manufacturer's instructions. The final product could be used immediately or stored at −20°C until further use.

All the RT-qPCR reactions were performed using the ViiA 7 Real-Time PCR machine (Applied BioSystems) according to the instruction of the manufacturer. For each target sample, amplifications were run in triplicate in 10 µL reaction volume containing 5 µL of 2x Power SYBR Green PCR Master Mix (Applied Biosystems, 4367659), 1.4 µL of cDNA, 0.5 µL of each primer at 2 µM, and 2.6 µL of nuclease-free water. The PCR program is as follows: denaturation at 95°C 10 min; 40 cycles of 95°C 15 s, 60°C 1 min; 1 cycle of 95°C 15 s, 60°C 1 min, 95°C 15 s. The last cycle provides the post PCR run melt curve, for assessment of the specifics of the amplification. Each couple of primers was designed in order to anneal on consecutive exons of the cDNA sequence, far from the exon-exon junction regions, except for the *Gapdh* gene. The size of amplicons varied between 84 bp and 177 bp. *Gapdh* housekeeping gene was used as internal control to normalize the variability in expression levels of each target gene, according to the $2^{-\Delta CT}$ method. For every experiment, data were further normalized to *Myh9* levels at the zygote stage.

Total RNA was extracted from 234, 159, 189, 152 embryos at zygote, four-cell, morula, and blastocyst stages, respectively, from six independent experiments.

| Gene name | Primer sequence forward | Primer sequence reverse |
| --- | --- | --- |
| *Myh9* (ex7–ex8) | CAATGGCTACATTGTTGGTGCC | AGTAGAAGATGTGGAAGGTCCG |
| *Myh10* (ex3–ex4) | GAGGGAAGAAACGCCATGAGA | GAATTGACTGGTCCTCACGAT |
| *Myh14* (ex25–ex26) | GAGCTCGAGGACACTCTGGATT | TTTCTTCAGCTCTGTCACCTCC |
| *Gapdh* (ex6–ex7) | CATACCAGGAAATGAGCTTG | ATGACATCAAGAAGGTGGTG |

## Bioinformatic analysis

Mouse and human single cell RNA sequencing data were extracted from *Deng et al., 2014* and *Yan et al., 2013*, analyzed as in *De Iaco et al., 2017*. In brief, single cell RNAseq datasets of human and mouse embryos (GSE36552, GSE45719) were downloaded and analyzed using the online platform Galaxy (usegalaxy.org). The reads were aligned to the reference genome using TopHat (Galaxy Version 2.1.1; *Kim et al., 2013*) and read counts generated with htseq-count (Galaxy Version 0.9.1; *Anders et al., 2015*). Normalized counts were determined with limma-voom (Galaxy Version 3.38.3 + galaxy3; *Law et al., 2014*).

## Microscopy

For live imaging, embryos were placed in 5 cm glass-bottom dishes (MatTek) under a Celldiscoverer 7 (Zeiss) equipped with a 20x/0.95 objective and an ORCA-Flash 4.0 camera (C11440, Hamamatsu) or a 506 axiovert (Zeiss) camera.

Embryos were imaged from E1.5 to E3.5 until the establishment of a stable blastocoel in WT control embryos. Using the experiment designer tool of ZEN (Zeiss), we set up a nested time-lapse in which all embryos were imaged every 30 min at three focal planes positioned 10 µm apart for 40–54 hr and each embryo was subjected to two 10 min-long acquisitions with an image taken every 5 s at two focal planes positioned 10 µm apart that were set 12 and 19 hr after the second cleavage division of a reference WT embryo. Embryos were kept in a humified atmosphere supplied with 5% $CO_2$ at 37°C. After imaging, mutant embryos were tracked individually for immunostaining and genotyping.

Live MYH9-GFP embryos were imaged at the zygote, four-cell, morula, and blastocyst stages. Live imaging was performed using an inverted Zeiss Observer Z1 microscope with a CSU-X1 spinning disc unit (Yokogawa). Excitation was achieved using 488 and 561 nm laser lines through a 63x/1.2 C Apo Korr water immersion objective. Emission was collected through 525/50 and 595/50 band pass filters onto an ORCA-Flash 4.0 camera (C11440, Hamamatsu). The microscope was equipped with an incubation chamber to keep the sample at 37°C and supply the atmosphere with 5% $CO_2$.

Immunostainings were imaged on the same microscope using 405, 488, 561, and 642 nm laser lines through a 63x/1.2 C Apo Korr water immersion objective; emission was collected through 450/50 nm, 525/50 nm, 595/50 band pass or 610 nm low pass filters.

## Data analysis

### Time-lapse of preimplantation development

Based on the time-lapses of preimplantation development from E1.5 to E3.5, we assessed the timing of the third, fourth, and fifth cleavage divisions as well as the dynamics of compaction and lumen growth following these definitions:

(1)Cleavages are defined as part of the same wave when they occur within 30 min of the cleavage of another cell within the same embryo. As embryos are recovered at E1.5, time-lapse imaging starts around the time of the second cleavage. As cell divisions are affected by the mutations of *Myh9* and/or *Myh10*, the cell number of recovered embryos is not necessarily equal to WT cell number. Therefore, we used the entire time-lapse to determine whether the first observed division corresponds to thesecond, third, fourth, or fifth wave. Failed cytokinesis is counted as part of cleavage waves. (2) Timing of maximal compaction is defined as the time when embryos stop increasing their contact angles. (3) Timing of blastocoel formation is taken as the time when a fluid compartment with a diameter of at least 20 µm is visible.

The timings of all cleavage divisions and morphogenetic events were normalized to the end of the third cleavage division (beginning of the 8-cell stage in WT embryos).

Surface contact angles were measured using the angle tool in Fiji. Only the contact angles formed by two adjacent blastomeres with their equatorial planes in focus were considered. For each embryo, between one and six contact angles were measured after completion of the third cleavage, just before the onset of the fourth cleavage, after completion of the fourth cleavage, and just before the onset of the fourth cleavage.

Lumen growth was assessed by an increase in the area of the embryo. Using FIJI, an ellipse was manually fitted around the embryo every hour starting from the time of blastocoel formation (defined here as when a lumen of at least 20 µm becomes visible) until the end of the time-lapse. Growth rates of individual embryos were calculated over a 7 hr time window following the time of blastocoel formation. Measured projected areas from different embryos were synchronized to the time of blastocoel formation and averaged. For a fused single-celled embryo, the mean blastocoel formation time of the control embryos from each experiment was calculated to synchronize the fused embryos (1 hr of deviation from the mean blastocoel formation time can be allowed to accommodate cell divisions and sampling time). For control embryos, only blastocyst growing by at least 35% of their projected area were considered.

The shape of the zona pellucida was measured by fitting an ellipse on the outer edge of the zona pellucida and measuring the long and short axes.

### Time-lapse of periodic contractions

Particle image velocimetry (PIV) analysis was performed using PIVlab 2.02 running on Matlab (*Thielicke and Stamhuis, 2020*; *Thielicke and Stamhuis, 2014*). Similar to *Maître et al., 2015*, time-lapse movies acquired every 5 s were processed using two successive passes through interrogation windows of 20/10 µm resulting in ~180 vectors per embryo. Vector velocities were then exported to Matlab for Fast Fourier Transform (FFT) analysis. The power spectrum of each embryo was then analyzed to assess the presence of a clear oscillation peak. The peak value between 50 and 120 s was taken as the amplitude, as this oscillation period range corresponds to the one where WT show oscillations (*Maître et al., 2015*; *Maître et al., 2016*). An embryo is considered as oscillating when the amplitude peaks 1.7 above background (taken as the mean value of the power spectrum signal of a given embryo) to determine whether it has a detectable oscillation. Power spectra of embryos from the same genotype were averaged.

### Time-lapse of Myh9-GFP embryos

Myh9-GFP intensity was measured in FIJI by fitting a circle with a radius of 50 µm over the sum projection of each embryo and measuring the mean gray value. Background intensity was measured in a

circular area with radius 5 μm and subtracted to the embryonic value. The corrected intensities were normalized to the zygote stage of maternal Myh9-GFP embryos.

## Immunostaining

We used FIJI to measure the levels of SOX2, CDX2, and YAP expression and localization by measuring the signal intensity of immunostainings. For each embryo, 15 cells (five ICM, five polar TE, five mural TE) were measured. In case an embryo had fewer than five ICM cells, a corresponding number of TE cells were added. If the embryo had fewer than 15 cells, all cells were measured. For each cell, the signal intensity was measured in a representative 3.7 μm$^2$ circular area of the equatorial nuclear plane and in a directly adjacent cytoplasmic area. Mitotic and apoptotic cells were excluded from analysis. We then calculated the nuclear to cytoplasmic ratio.

To obtain apico-basal intensity profiles of polarity markers, we selected confocal slices cutting through the equatorial plane of a surface cell lining the blastocoel. We drew a 15-μm-long and 0.4-μm-thick line from the cell-medium interface through the cell into the cell-lumen or cell-vacuole interface. Lowest/highest intensity values were normalized to 0 and 1, respectively.

To count cell number, we used DAPI to detect nuclei and phalloidin staining to detect cells. Cells were considered outer cells if they were in contact with the outside medium.

To measure the 3D properties of blastocysts, we manually segmented the surface of the embryo and of the blastocoel using Bitplane Imaris. Volumes and ellipticity were obtained from Imaris. The cellular volume was obtained by subtracting the blastocoel volume from the total volume of the embryo.

## Statistics

Mean, standard deviation, SEM, lower and upper quartiles, median, Pearson's correlation coefficients, unpaired two-tailed Welch's $t$ test, Mann-Whitney $U$ test, and Pearson's Chi$^2$ test with Yates's correction for continuity p values were calculated using Excel (Microsoft) and R (http://www.r-project.org). Pearson's correlation statistical significance was determined on the corresponding table. Statistical significance was considered when $p<10^{-2}$. Boxplots show the median (line), interquartile range (box), 1.5 x interquartile range (whiskers), and remaining outliers (dots).

The sample size was not predetermined and simply results from the repetition of experiments. No sample was excluded. No randomization method was used. The investigators were not blinded during experiments.

## Acknowledgements

We thank the imaging platform of the Genetics and Developmental Biology unit at the Institut Curie (PICT-IBiSA@BDD), member of the French National Research Infrastructure France-BioImaging (ANR-10-INBS-04) for their outstanding support; the animal facility of the Institut Curie for their invaluable help. We thank Aurélie Teissendier for help with bioinformatic analysis, Victoire Cachoux for help with image analysis, and Yohanns Bellaïche for discussion and advice on the manuscript. Research in the lab of J-LM is supported by the Institut Curie, the Centre National de la Recherche Scientifique (CNRS), the Institut National de la Santé Et de la Recherche Médicale (INSERM), and is funded by grants from the ATIP-Avenir program, the Fondation Schlumberger pour l'Éducation et la Recherche via the Fondation pour la Recherche Médicale, the European Research Council Starting Grant ERC-2017-StG 757557, the European Molecular Biology Organization Young Investigator program (EMBO YIP), the INSERM transversal program Human Development Cell Atlas (HuDeCA), Paris Sciences Lettres (PSL) 'nouvelle équipe' and QLife (17-CONV-0005) grants and Labex DEEP (ANR-11-LABX-0044), which are part of the IDEX PSL (ANR-10-IDEX-0001–02). MFS is funded by a Convention Industrielle de Formation pour la Recherche (No 2019/0253) between the Agence Nationale de la Recherche and Carl Zeiss SAS. ÖÖ is funded from the European Union's Horizon 2020 research and innovation program under the Marie Skłodowska-Curie grant agreement No 666003 and benefits from the EMBO YIP bridging fund.

## Additional information

### Competing interests

Markus Frederik Schliffka: is employed by Carl Zeiss SAS via a public PhD programme Conventions Industrielles de Formation par la Recherche (CIFRE) co-funded by the Association Nationale de la Recherche et de la Technologie (ANRT). The other authors declare that no competing interests exist.

### Funding

| Funder | Grant reference number | Author |
|---|---|---|
| Association Nationale de la Recherche et de la Technologie | 2019/0253 | Markus Frederik Schliffka |
| H2020 Marie Skłodowska-Curie Actions | 666003 | Özge Özgüç |
| Fondation pour la Recherche Médicale | | Özge Özgüç |
| Centre National de la Recherche Scientifique | | Jean-Léon Maître |
| H2020 European Research Council | ERC-2017-StG 757557 | Jean-Léon Maître |
| Schlumberger Foundation | | Jean-Léon Maître |
| Université de Recherche Paris Sciences et Lettres | 17-CONV-0005 | Jean-Léon Maître |
| Agence Nationale de la Recherche | ANR-11-LABX-0044 | Jean-Léon Maître |
| Institut National de la Santé et de la Recherche Médicale | HuDeCA | Jean-Léon Maître |
| European Molecular Biology Organization | Young Investigator Programme | Jean-Léon Maître |
| Agence Nationale de la Recherche | ANR-10-IDEX-0001-02 | Jean-Léon Maître |

The funders had no role in study design, data collection and interpretation, or the decision to submit the work for publication.

### Author contributions

Markus Frederik Schliffka, Anna Francesca Tortorelli, Conceptualization, Resources, Data curation, Formal analysis, Validation, Investigation, Visualization, Methodology, Writing - review and editing; Özge Özgüç, Data curation, Software, Formal analysis, Visualization, Methodology, Writing - review and editing; Ludmilla de Plater, Resources, Methodology; Oliver Polzer, Formal analysis, Visualization; Diane Pelzer, Software, Formal analysis, Visualization, Writing - review and editing; Jean-Léon Maître, Conceptualization, Resources, Data curation, Software, Formal analysis, Supervision, Funding acquisition, Validation, Investigation, Visualization, Methodology, Writing - original draft, Project administration, Writing - review and editing

### Author ORCIDs

Markus Frederik Schliffka ![ORCID] https://orcid.org/0000-0002-5128-1653
Anna Francesca Tortorelli ![ORCID] https://orcid.org/0000-0002-9995-9582
Özge Özgüç ![ORCID] http://orcid.org/0000-0002-1545-1715
Ludmilla de Plater ![ORCID] http://orcid.org/0000-0002-0982-5960
Oliver Polzer ![ORCID] http://orcid.org/0000-0003-4970-6058

Diane Pelzer (iD) http://orcid.org/0000-0001-6906-2451
Jean-Léon Maître (iD) https://orcid.org/0000-0002-3688-1474

## Ethics

Animal experimentation: All animal work is performed in the animal facility at the Institut Curie, with permission by the institutional veterinarian overseeing the operation (APAFIS #11054-2017082914226001). The animal facilities are operated according to international animal welfare rules.

## Decision letter and Author response

Decision letter https://doi.org/10.7554/eLife.68536.sa1
Author response https://doi.org/10.7554/eLife.68536.sa2

## Additional files

### Supplementary files

• Transparent reporting form

### Data availability

The microscopy data, ROI and analyses are available on the following repository under a CC BY-NC-SA license: https://ressources.curie.fr/mzmyh/.

The following previously published datasets were used:

| Author(s) | Year | Dataset title | Dataset URL | Database and Identifier |
|---|---|---|---|---|
| Deng Q, Ramsköld D, Reinius B, Sandberg R | 2014 | Single-cell RNA-Seq reveals dynamic, random monoallelic gene expression in mammalian cells | https://www.ncbi.nlm.nih.gov/geo/query/acc.cgi?acc=GSE45719 | NCBI Gene Expression Omnibus, GSE45719 |
| Tang F, Qiao J, Li R | 2013 | Tracing pluripotency of human early embryos and embryonic stem cells by single cell RNA-seq | https://www.ncbi.nlm.nih.gov/geo/query/acc.cgi?acc=GSE36552 | NCBI Gene Expression Omnibus, GSE36552 |

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

# Appendix 1

**Appendix 1—table 1.** Statistical analysis of compaction data.
Mean contact angles and developmental times related to *Figure 1B* and *Figure 3B* with associated SEM, embryo number, and p value from unpaired two-tailed Welch's *t* test against WT.

| | | Time after last third cleavage | | | | Time before intial fourth cleavage | | | | Time after last fourth cleavage | | | | Time before intial fifth cleavage | | | |
|---|---|---|---|---|---|---|---|---|---|---|---|---|---|---|---|---|---|
| | | Mean | SEM | Embryos | P value | Mean | SEM | Embryos | p value | Mean | SEM | Embryos | p value | Mean | SEM | Embryos | P value |
| WT | Time after 3rd cleavage [h] | 0 | 0 | 23 | NA | 7.0 | 0.3 | 23 | NA | 10.9 | 0.3 | 21 | NA | 16.7 | 0.4 | 22 | NA |
| | Contact angle [°] | 87 | 3 | | NA | 147 | 2 | | NA | 140 | 3 | | NA | 148 | 2 | | NA |
| mzMyh9 | Time after 3rd cleavage [h] | 0 | 0 | 15 | NA | 9.8 | 0.5 | 15 | 0.0001 | 13.8 | 0.6 | 10 | 0.002 | 18.8 | 0.5 | 8 | 0.01 |
| | Contact angle [°] | 85 | 2 | | 0.5 | 125 | 4 | | 0.00001 | 104 | 5 | | 0.00001 | 131 | 8 | | 0.02 |
| mMyh9 | Time after 3rd cleavage [h] | 0 | 0 | 8 | NA | 10.4 | 0.7 | 8 | 0.001 | 14.6 | 0.8 | 8 | 0.002 | 18.7 | 0.7 | 3 | 0.002 |
| | Contact angle [°] | 86 | 5 | | 0.5 | 115 | 6 | | 0.0001 | 112 | 7 | | 0.003 | 113 | 21 | | 0.1 |
| mzMyh10 | Time after 3rd cleavage [h] | 0 | 0 | 11 | NA | 5.4 | 0.4 | 11 | 0.004 | 10.2 | 0.6 | 11 | 0.2 | 15.8 | 0.4 | 11 | 0.1 |
| | Contact angle [°] | 87 | 5 | | 0.8 | 121 | 4 | | 0.0001 | 126 | 5 | | 0.03 | 148 | 4 | | 0.8 |
| mMyh10 | Time after 3rd cleavage [h] | 0 | 0 | 20 | NA | 6.3 | 0.4 | 20 | 0.2 | 10.8 | 0.3 | 20 | 0.5 | 16.8 | 0.6 | 20 | 0.9 |
| | Contact angle [°] | 85 | 3 | | 0.3 | 128 | 5 | | 0.0004 | 128 | 3 | | 0.006 | 149 | 2.8 | | 0.5 |
| mzMyh9; mzMyh10 | Time after 3rd cleavage [h] | 0 | 0 | 7 | NA | 9.7 | 0.8 | 3 | 0.07 | 12 | NA | 1 | NA | NA | | | |
| | Contact angle [°] | 89 | 6 | | 0.8 | 117 | 4 | | 0.006 | 84.5 | NA | | NA | | | | |
| mzMyh9; mMyh10 | Time after 3rd cleavage [h] | 0 | 0 | 7 | NA | 6.9 | 1.7 | 4 | 1.0 | 9 | 0.5 | 2 | 0.1 | 12 | NA | 1 | NA |
| | Contact angle [°] | 83 | 5 | | 0.6 | 100 | 5 | | 0.002 | 108 | 14 | | 0.07 | 113 | NA | | NA |
| mMyh9; mMyh10 | Time after 3rd cleavage [h] | 0 | 0 | 7 | NA | 8 | 1.3 | 5 | 0.5 | 11 | 0.5 | 3 | 1.0 | NA | | | |
| | Contact angle [°] | 76 | 7 | | 0.1 | 110 | 5 | | 0.0006 | 105 | 3 | | 0.002 | | | | |

**Appendix 1—table 2.** Statistical analysis of cell cycle duration data.
Mean durations of 8-cell stage, fourth wave of cleavage, and 16-cell stage related to *Figures 2B* and *5B* with associated SEM, embryo number, and p value from unpaired two-tailed Welch's *t* test against WT.

| | Duration of 8-cell stage [h] | | | | Duration of fourth wave of cleavages [h] | | | | Duration of 16-cell stage [h] | | | |
|---|---|---|---|---|---|---|---|---|---|---|---|---|
| | Mean | SEM | Embryos | p value | Mean | SEM | Embryos | p value | Mean | SEM | Embryos | p value |
| WT | 7.0 | 0.3 | 23 | NA | 4.0 | 0.3 | 23 | NA | 5.5 | 0.4 | 23 | NA |
| mzMyh9 | 9.8 | 0.5 | 15 | 0.0001 | 4.8 | 0.7 | 15 | 0.3 | 5.7 | 0.8 | 14 | 0.8 |
| mMyh9 | 10.4 | 0.7 | 8 | 0.001 | 4.2 | 1.2 | 8 | 0.9 | 7.0 | 1.1 | 8 | 0.1 |
| mzMyh10 | 5.4 | 0.4 | 11 | 0.004 | 4.8 | 0.8 | 11 | 0.2 | 5.6 | 0.5 | 11 | 0.9 |
| mMyh10 | 6.3 | 0.4 | 20 | 0.2 | 4.5 | 0.4 | 20 | 0.3 | 6.1 | 0.5 | 20 | 0.4 |

*Continued on next page*

*Appendix 1—table 2 continued*

| | Duration of 8-cell stage [h] | | | | Duration of fourth wave of cleavages [h] | | | | Duration of 16-cell stage [h] | | | |
|---|---|---|---|---|---|---|---|---|---|---|---|---|
| | Mean | SEM | Embryos | p value | Mean | SEM | Embryos | p value | Mean | SEM | Embryos | p value |
| mzMyh9; mzMyh10 | 10.3 | 1.4 | 8 | 0.06 | 2.0 | 0.8 | 3 | 0.1 | | | | |
| mzMyh9; mMyh10 | 8.1 | 1.4 | 6 | 0.5 | 4.3 | 1.1 | 3 | 0.8 | NA | | | |
| mMyh9;mMyh10 | 9.3 | 1.0 | 8 | 0.06 | 3.6 | 0.7 | 5 | 0.6 | | | | |

**Appendix 1—table 3.** Statistical analysis of periodic contraction data.

Mean amplitude of periodic movements and contact angle related to *Figure 2H* with associated SEM, embryo number, and p value from Mann-Whitney *U* test against WT.

| | | Maximum amplitude within 50–120 s oscillation period range [μm] | | | Contact angle [°] | | |
|---|---|---|---|---|---|---|---|
| | Embryos | Mean | SEM | p value | Mean | SEM | p value |
| WT | 21 | 0.146 | 0.009 | NA | 126 | 5 | NA |
| mzMyh9 | 15 | 0.037 | 0.002 | 0.000003 | 78 | 3 | 0.000003 |
| mMyh9 | 8 | 0.041 | 0.003 | 0.00002 | 82 | 6 | 0.001 |
| mzMyh10 | 11 | 0.077 | 0.009 | 0.004 | 103 | 6 | 0.02 |
| mMyh10 | 20 | 0.094 | 0.014 | 0.007 | 108 | 5 | 0.02 |

