## [Decision Letter]

[Editors' note: this paper was reviewed by Review Commons.]

**Acceptance summary:**

Your study provides novel information regarding the role of Myh9 and Myh10 in preimplantation mammalian development. We feel that such information will lead to a better understanding to the initial steps in early embryo formation and cell fate specification.

---

## [Author Response]

Reviewer #1 (Evidence, reproducibility and clarity (Required)):Summary:Previously, the authors showed the importance of contractile force in cell positioning and cell fate specification in preimplantation mouse development. In this study, the authors generated maternal-zygotic mutants of the nonmuscle myosin-II heavy chain (NMHC) genes Myh9 and Myh10, and quantitatively analyzed their development using time-lapse microscopy and immunostaining. The authors first examined the expression of NMHCs. Myh9 and Mhy10 are present in preimplantation embryos, and Myh9 is maternally inherited. Single maternal-zygotic mutants of Myh9 or Myh10 revealed that maternal Myh9 plays a major role in actomyosin contractility. In maternal Myh9 mutants, compaction and contractility at the 8-cell stage were reduced. Maternal Myh9 mutants demonstrated a longer 8-cell stage, and mutant blastocysts had reduced cell numbers. Cell positioning was not affected; however, cell differentiation was slightly affected by reduced expression of TE and ICM markers. Maternal Myh9 mutants formed blastocoels, but lumen opening was observed earlier than that in wild-type embryos. In double maternal-zygotic mutants of Myh9 and Myh10, cytokinesis was severely affected.Nevertheless, TE fate was specified and embryos formed blastocoels. Interestingly, single-celled mutants swelled upon the formation of fluid-filled vacuoles in their cytoplasm. Similar TE fate specifications and cytoplasmic vacuoles were also observed with single-celled embryos produced by blastomere fusion. Based on these results, the authors concluded that maternal Myh9 is the major NMHC. However, Myh10 can significantly compensate for the loss of Myh9, and that cell fate specification and morphogenesis are independent of the success of cell division.Minor comments:Overall, the conclusions of this study are supported by high-quality data. However, I have a few minor concerns:

We thank the referee for her/his careful analysis of our manuscript.

1. Line 200~205. The authors showed the correlation between the cell number at the blastocyst stage and the 8-cell stage, and concluded that "the lengthened 8-cell stage of mzMyh9 is an important determinant of their reduced cell number at the blastocyst stage". This conclusion is not well supported because of several reasons. First, the timing of cell count is not clear. Cell number was compared at the blastocyst stage, but Figure 1c shows that mzMyh9 embryos initiate blastocoel formation earlier than wild-type embryos. Therefore, if cell count timing was determined based on the blastocyst morphology of the embryos, the timing of cell count (i.e., time after 3rd cleavage) for mzMyh9 mutants is earlier than that observed for wild-type embryos. This shorter culture time likely contributes to the reduced cell number of mzMyh9. Second, the authors only showed a correlation, and no experimental data supporting this conclusion were shown. If the cell number was counted at the same time after the 3rd cleavage, and if the authors' hypothesis is correct, then culturing mzMyh9 mutants for an additional three hour, which is the difference in the duration of the 8-cell stage, should make the cell numbers of mutants comparable to those of wild-type blastocysts.

Although, this correlation provides the best explanation we had based on the data, we agree that the statement above is weakly supported by our study. We do not want to make a strong point about it since we do not think it brings much to the narrative of the study. We have removed the sentence.

2. Discussion. In the paragraph starting from line 405, the authors discussed the inconsistencies in the observation of the phenotypes of mzMyh9 and mzMyh10 mutants with the conclusions of previous studies by others about cell polarization. It will be informative to also discuss about inconsistency with their previous observations on cell fate. In their previous report (reference 8), the authors concluded that without contractile forces, blastomeres adopt an inner-cell-like fate regardless of their position. This is clearly opposite of the phenotype of mzMyh9;mzMyh10 mutants, in which all the cells are specified to TE. Please add a discussion addressing this discrepancy.

The data provided here are consistent with the ones from ref 8 (Maître *et al.,* 2016): reduced contractility (*Myh9* KO, double *Myh9;Myh10* KO or Blebbistatin treatment) leads to reduced CDX2 levels. In ref 8, CDX2 and YAP are checked at the 16-cell stage, before the definitive differentiation into TE and ICM, whereas here we present data at the mid-blastocyst stage (~64 cells). We had not checked *SOX2* in ref 8 since it is not expressed at such early stage, so we cannot conclude about this marker.

We want to clarify that, as stated in the manuscript, in *mzMyh9;mzMyh10* KO we detect CDX2 in 5/7 embryos only and therefore not all cells are correctly specified into TE. However, *SOX2* could be detected in the inner cell of the one embryo that produced an inner cell. We had not discussed this issue further since it is difficult to conclude much from such rare events and we would prefer to keep it as such.

To strengthen our argument about reduced differentiation in NMHC mutant embryos, we now provide YAP immunostaining (Figure S4). YAP is correctly patterned in *Myh10* mutants and shows slightly less defined nuclear localization in *Myh9* mutants, in agreement with our previous observations on CDX2 in the present study and previous observations on YAP at the 16-cell stage (Maître et al. 2016).

Together, we can conclude that, at the 16-cell stage, when ICM fate is not engaged yet (no detectable *SOX2* expression), “inhibition of contractility causes (…) blastomeres to become inner-cell-like with respect to (…) Yap localization and Cdx2 levels, despite their external position” (Maître *et al.,* 2016). At the blastocyst stage embryos with chronically impaired contractility can succeed in some but not all cases to produce TE (this study). Between these two developmental stages, blastomeres are exposed to prolonged signals from the apical domain and can be strongly deformed by the growing lumen. Based on the literature (Hirate et al. 2013, Dupont et al. 2011), both of these stimuli could potentially favor YAP nuclear localisation despite low contractility.

3. Throughout the paper, the description of gene and protein symbols should follow the rules of MGI's guidelines for nomenclature of genes (http://www.informatics.jax.org/mgihome/nomen/gene.shtml#gene_sym). Gene and allele symbols are italicized. Protein symbols use all uppercase letters and are not italicized.

We have corrected this.

4. Line 163. The term "contact angles" are used without any explanation or definition. The term should be introduced with a brief explanation in the text, preferably with a figure. It should help facilitate the understanding of the scientists working in different fields.

We have labelled a contact angle on Figure 1A and specified this in the text and in the figure legend.

Reviewer #1 (Significance (Required)):The importance of actomyosin contractility in compaction, cell polarization, cell positioning, and cell fate specification in preimplantation embryos has been reported by several groups, mostly using chemical inhibitors, except for the study cited in reference 8, in which chimeras of wild-type and mMyh9 mutant embryos were used. This is the first genetic analysis of the roles of actomyosin contractility in the development of preimplantation embryos. Thus, the major advancement of this study is the genetic dissection of the roles of actomyosin contractility in preimplantation mouse development, and clarifying the contribution of maternal/zygotic Myh9 and Myh10 genes. While the phenotypes of reduced compaction and blastomere contractility are consistent with those observed in previous studies, polarization and TE fate specification of the mutant cells appear inconsistent with the conclusions of previous inhibitor experiments, which show defects in polarization processes and fate specification to ICM. These are potentially important issues, but detailed analyses were not performed. The requirement of actomyosin contractility for the cytokinesis of preimplantation embryos is also a novel finding, although it is expected from studies conducted in other systems. Vacuole formation in single-celled mzMyh9;mzMyh10 mutants in a timely manner suggested that fluid accumulation is a cell autonomous process and that cell differentiation occurs independently of cell division. These are also novel findings, although the latter is somewhat expected from previous studies performed using cell number manipulated embryos.In summary, the conceptual advance offered by this study is small. However, this is a high-quality study and makes critical observations in the field of preimplantation mouse development. Scientists in the field of developmental biology, especially those working on preimplantation development, should be interested in this paper.My field of expertise is preimplantation development.

We thank the reviewer for her/his appreciation of our work. We want to argue that we did perform a very detailed analysis of the development of the NMHC mutant embryos, with multiple quantitative image and data analyses to thoroughly and objectively characterise the phenotypes of these mutants. If by “detailed analysis”, the reviewer meant a molecular dissection of the phenotype, we argue that 1/ checking the end result (i.e. presence of TE and ICM markers, presence of polarised fluid transport) was sufficient to assess the functionality of biological processes without checking every steps of a signalling cascade; 2/ we now provide additional molecular information on the state of YAP and apico-basal polarisation (Figure S3-4).

Reviewer #2 (Evidence, reproducibility and clarity (Required)):In this manuscript, Schliffka et al. report that maternally deposited Myh9 is the major NMHC in preimplantation embryonic morphogenesis and complete removal of both Myh9 and Myh10 caused severe cytokinesis failure similar to tissue culture cells. Interestingly, although the mutant embryos completely failed cytokinesis thus forming single-celled embryos, they initiated trophoblast gene expression and vacuolization (likely similar to blastocoel formation), suggesting that the timing of preimplantation developmental events is independent from cell number and morphogenetic events.

We thank the reviewer for her/his appreciation of our work.

Major commentsVacuolization in single-celled embryos is interesting. In the images, there looks to be two types of vacuoles, Factin positive and negative. The authors speculate the similarity to blastocoel formation. To support this, it is important to stain them with some basolateral markers like Na^+^ ATPase, E-cadherin and B-catenin. It is also important to confirm if the apical domain is properly formed by staining the apical domain markers like aPKC and Pard6.

We thank the reviewer for this suggestion. We now provide immunostaining of single *Myh9* or *Myh10* and double *Myh9;Myh10* mutants for aPKC (PRKCz), Na/K ATPase (ATP1A1), Aquaporin-3 (AQP3), the best basolateral marker in our hands, which is also very relevant to fluid pumping, CDH1 and F-actin (Figure S3). We observe that these markers localise similarly in multiple-celled and single-celled embryos, suggesting that vacuoles de facto substitute for the basolateral compartment normally consisting of cell-cell contacts and the lumen. This suggests that the same machinery is at the origin of the fluid inside the lumen and inside vacuoles.

Minor commentsAll gene names should be Italicized.

We have corrected this.

L157. Myh10 and Myh9 should be mMyh10 and mMyh9.

We have corrected this.

L294 1/8 embryos. What does this mean?

This means this was observed in 1 embryo out of 8 in total.

L333 6/25 embryos. Does this mean 6 out of 25 embryos combined all maternal double mutants?

Precisely.

L438-442. I do not find these embryos are similar to tetraploid embryos. I suggest to remove the sentences.

We have removed the sentences.

Reviewer #3 (Evidence, reproducibility and clarity (Required)):This study investigates the roles of non-muscle myosin in development, reporting a requirement for maternal and zygotic Mhy9 and 10. Strengths of the study include robust genetic techniques, innovative nested imaging to visualize events over different timescales within the same embryos, and analysis of morphological as well as transcriptional/cell fate phenotypes. However, the somewhat superficial phenotype analysis limits the authors' ability to draw strong mechanistic conclusions about what is going on in these mutants. Is cell polarization normal? Is cell signaling (HIPPO signalling) normal?

We thank the reviewer for carefully assessing our study. We argue that we have thoroughly characterized the phenotypes of the NMHC mutants, which allowed us to draw many important mechanistic conclusions (such as the ability of NMHC mutants to polarise, or to pump fluid in a cell autonomous manner). Each mutant embryo has been imaged at multiple time scales, stained and genotyped. The time-lapses and immunostaining have been extensively quantified using manual as well as automated methods such as particle image velocimetry. We also provided fusion experiments, which phenocopy some aspects of the mutants to provide evidence of the mechanisms causing the observed phenotype.

Nevertheless, we agree that one can always do more and that we had focused on the biological processes (lineage specification, morphogenesis and cleavages) rather than molecular characterisation. Although polarised fluid pumping ascertains a functioning epithelial polarity, we now provide immunostaining of polarity markers in mutant embryos. Although CDX2 and *SOX2* staining inform on the output of the signalling cascade leading to effective TE and ICM differentiation, we now provide YAP immunostaining of mutant embryos. We hope this satisfies the request from the reviewer.

What determines whether an embryo can form an inside cell or not?

This is an outstanding question. Cells can internalise by oriented cell division or contractility mediated cell sorting. Contractility-mediated internalisation functions with only 2 cells (as when doublets of 16-cell stage blastomeres form a cell-in-a-cell structure) but requires to grow above a tension asymmetry threshold (of 1.5 in WT and most likely above 3 in these mutants due to their poor compaction, see Maître et al. 2016). Oriented cell division only works if there is a cell-cell contact to push dividing cells in between. Therefore, at least 3 cells are required for an inner cell to be internalised by this mechanism.

In double mutants, the average cell number is 2.9. No embryo consisting of only 2 cells contained an inner cell, about half of embryos with 3-5 cells contained a single inner cell and all embryos with 6 cells or more contained inner cells (Figure 4D). Based on the low contractility of double mutants, we can speculate that they do not succeed in overcoming the tension asymmetry threshold. This would explain why no inner cell is observed in embryos with only 2 cells. We can speculate that with 3-5 cells, oriented divisions could occur thanks to the presence of functional polarity (Korotkevitch et al. 2017). We have added a discussion about this important matter.

Similarly, the manuscript would benefit from rewriting to reframe the authors' discoveries within the context of what is known regarding lineage specification (e.g., why does CDX2/SOX2 expression indicate normal lineage specification). Additional minor comments are listed below.

We elaborate on these points.

Minor comments:Introduction focuses overly on the work of the π and his mentor, giving the presentation an unnecessarily biased quality.

We have corrected this to the best of our ability. Please note that, to our knowledge, there are 8 studies (Anani et al., 2014; Maître et al., 2015; Samarage et al., 2015; Maître et al., 2016; Zhu et al., 2017; Zenker et al., 2018; Chan et al., 2019; Dumortier et al., 2019) looking in more or less details into the contractility of the preimplantation embryo. We mention and cite all of these studies.

The text asserts that Myh9 levels are highest during zygote stage, on the basis of qPCR (Figure S1A), and that this is also observed by RNA-seq (Figure S1B). However, this conclusion is not supported by the data shown.

We have corrected this.

Would be nice to repeat the qPCR on the mz null.

We agree with the referee that this would help in assessing the level of compensation between NMHC paralogs in individual mutants. Our qPCR protocol requires a few tens of embryos to be able to amplify the different paralogs.

Unfortunately, pooling embryos from our current mating strategy would result in pooling homozygote and heterozygote mutants as we cannot know a priori which embryo is of which genotype.

We believe that, as nice as this information would be, the current study does not require this information, which would be technically challenging.

Were the measurements shown in Figure S1F taken from the images shown in Figure S1E? If so, the authors should clarify how the measurements were normalized, since the images in Figure S1E were clearly taken with different camera settings (as judged by background fluorescence level surrounding the embryos).

The camera settings were identical but the LUT are set differently (to the maximal signal of a given genotype) so that some signal is visible. The signal intensities are so different between genotypes that if set to a common LUT, we either get the maternal GFP as a saturated white circle or the other genotypes as black images. We explain our LUT settings both in the methods and figure legends.

As an alternative to the current data presentation, we would be fine to have the same LUT for all images and show almost black images for WT and paternal GFP.

Can't really conclude that Myh9 is essential for compaction since compaction occurs (albeit abnormally) in the absence of Myh9 (line 177-178).

Our statement is “we conclude that maternal *Myh9* is essential for embryos to compact fully”.

WT and *mzMyh10* mutants increase their contact angles by 60° whereas *mzMyh9* only grow by 30°. Double mutants compact less than single *Myh9* mutants. Therefore, the compaction movement is halved in *mzMyh9* and the residual weak compaction could be explained by compensation from *Myh10*. We stand by our statement.

ine 211: "observe" rather than "measure".

We have corrected this.

If the embryos achieve proper ICM/TE ratio, in spite of having half the number of cells in the mutants, is that to be expected? Would/do halved embryos also possess the same ICM/TE ratio? Or is this outcome peculiar to the mutants?

This is an interesting question on which we had not sufficiently elaborated. Our experiments with cell fusion at the 4-cell-stage (Figure S5) produced embryos with reduced cell number. These resemble *Myh9* mutant embryos in the aspect that they show a reduced cell number while maintaining the total embryonic cell mass. In both cases, the ICM/total cell ratio is similar to control embryos. This indicates a robust mechanism of ICM/TE ratio setting that is robust to the cell number change observed in the single mutant. We have added a discussion about this.

Line 222: what is the evidence that Cdx2 and Sox2 are TE and ICM markers?

We have added references to the studies from Strumpf et al., 2005 and Avilion et al., 2003 to support these claims.

Is the reported reduction in CDX2 and SOX2 levels due to a stage-delay? What would the comparison look like in wt embryos with half as many cells? Timing of lumen formation may or may not indicate developmental timing…

We address this point by fusing embryos to half the cell number and find that the fate marker levels are specifically affected as a result of mutation of *Myh9* (Figure S5).

We agree that the timing of lumen formation is unlikely to be a good reference for staging and we did not use this event. We do synchronise embryos based on lumen opening only when comparing lumen growth rate.

Line 240 – what was the correction on the multiple pairwise comparisons (multiple t tests)?

To compare lumen growth rate, individual growth rates of mutants are compared to those of WT using Student’s t test. Growth rates are considered as normally distributed and independent (not pairwise).

Lumen forms on time in mutants, despite having fewer cells. Alternatively, lumen forms early, prior to acquisition of proper cell number. Is there a reason the authors did not consider this alternative?

The referee is correct. Lumens form with fewer cells in mutant embryos and therefore prior to the acquisition of proper cell number.

Lines 306 and 339: why does lack of SOX2 expression suggest that the lineage specification program is intact?Why does expression of CDX2 suggest TE initiation has occurred normally? The regulation of these two markers was not introduced.

We have better introduced and justified this aspect.

Line 349: why is blastocoel formation a cell-autonomous property when it clearly occurs extracellularly? Does this also happen in wild type embryos?

Blastocoel formation is clearly a multi-cellular process. We argue that fluid accumulation is not. The implications for WT embryos are that fluid can be accumulated in the blastocoel entirely trans-cellularly (no need for fluid to flow through cell-cell junction).

Speculate in Discussion on why the ML-7/Blebbistatin experiments results could differ from the genetic results produced here.

Blebbistatin experiments are in agreement with the mutant data. ML-7 experiments are partially in agreement with the mutant data. The discrepancy lies in the effect on cell polarity. ML-7 affects kinases other than the MLCK, such as PKC, which is a known regulator of cell polarity during preimplantation development. Although this is speculative, we specify this in the revised manuscript.

Can these mutant embryos implant?

We grow colonies of heterozygous mutants, therefore *mMyh9*, *mMyh10* and *mMyh9*;*mMyh10* embryos are viable and must be able to implant. As for homozygous mutants, they are not viable and we do not know whether they can implant.

Reviewer #3 (Significance (Required)):The study provides the first strong evidence of a requirement for non-muscle myosin in epithelialization. This is significant to embryology and to epithelial biology.

We thank the reviewer for appreciating the significance of our study. We want to clarify that our study provides evidence for NMHC as NOT being required for de novo epithelialization.